# C-terminal modification and functionalization of proteins via a self-cleavage tag triggered by a small molecule

Yue Zeng[1,2,5], Wei Shi[1,5], Zhi Liu [1,3], Hao Xu[3], Liya Liu[1], Jiaying Hang[1], Yongqin Wang[4], Mengru Lu[4], Wei Zhou[4], Wei Huang [1,2,3,4] & Feng Tang [1,2,4] ✉

The precise modification or functionalization of the protein C-terminus is essential but full of challenges. Herein, a chemical approach to modify the C-terminus is developed by fusing a cysteine protease domain on the C-terminus of the protein of interest, which could achieve the non-enzymatic C-terminal functionalization by InsP$_6$-triggered cysteine protease domain self-cleavage. This method demonstrates a highly efficient way to achieve protein C-terminal functionalization and is compatible with a wide range of amine-containing molecules and proteins. Additionally, a reversible C-terminal de-functionalization is found by incubating the C-terminal modified proteins with cysteine protease domain and InsP$_6$, providing a tool for protein functionalization and de-functionalization. Last, various applications of protein C-terminal functionalization are provided in this work, as demonstrated by the site-specific assembly of nanobody drug conjugates, the construction of a bifunctional antibody, the C-terminal fluorescent labeling, and the C-terminal transpeptidation and glycosylation.

Chemical modification of proteins has emerged as a versatile strategy for the gain of functions, such as increasing the half-life[1], labeling the target receptor[2], modulating protein-protein interactions[3], etc. Great efforts have been made to site-specifically modify the proteins of interest (POIs)[4,5], including N- or C-terminal modification[6,7], the side-chain of amino acid[8], such as amine (lysine)[9], sulfhydryl (cysteine)[10,11], imidazole (histidine)[12,13] and phenol (tyrosine)[14]. Despite this, site-specific functionalization of proteins, particularly for C-terminus, remains a great challenge suffering from the multiple reactive sites on proteins. Recently, chemo-enzymatic C-terminal modification of the POIs has been widely studied[7,15]. This involves fusing the C-terminus with unique tags that can be "decorated" by the corresponding enzymes, such as sortase A[16–18], trypsiligase[19], tubulin tyrosine ligase (TTL)[20], asparaginyl endopeptidases (AEPs)[21–26], PAM15[27],

carboxypeptidase Y (CPaseY)[28], and the combination of multiple enzymes[29]. The chemical approach for C-terminal modification is based on the difference in oxidation potential between terminal carboxylic acids and in-chain Glu or Asp residues with a decarboxylative photoredox methodology[30]. In addition, the split intein system has been well-developed in the past two decades, especially by Tom W. Muir[31–33], to achieve the semi-synthesis or N/C-terminal modification of proteins, and has become an important chemical tool to study protein functions. The intein domain can self-split or be removed by MESNA (sodium 2-sulfanylethanesulfonate) and give a thioester intermediate which could further react with an N-Cys-containing protein/peptide or small molecule. Besides, Thom et al. employed the intein system to achieve hydrazinolysis of POI C-terminus by simply incubating POI-intein with hydrazide and then following up with a chemoselective

[1]State Key Laboratory of Drug Research, Center for Biotherapeutics Discovery Research, Shanghai Institute of Materia Medica, Chinese Academy of Sciences, No.555 Zuchongzhi Rd, Pudong, Shanghai 201203, China. [2]University of Chinese Academy of Sciences, No.19A Yuquan Road, Beijing 100049, China. [3]School of Chinese Materia Medica, Nanjing University of Chinese Medicine, No. 138 Xianlin Rd, Nanjing 210023, China. [4]School of Pharmaceutical Science and Technology, Hangzhou Institute of Advanced Study, Hangzhou 310024, China. [5]These authors contributed equally: Yue Zeng, Wei Shi. ✉e-mail: tangfeng2013@simm.ac.cn

reaction with ketone or aldehyde containing moieties[34]. Meng-Jung Chiang et al. also constructed an Fc Domain protein-small molecule conjugate by fusing intein onto the C-terminus of Fc, which was converted into an Fc-thioester by MESNA and reacted with a small molecule containing N-Cys-linker[35]. Another similar approach was developed by Qiao et al. that the POI is fused with a Cys on the C-terminus and is activated by 2-nitro-5-thiocyanatobenzoic acid (NTCB), which generates an intermediate that can be attacked by a subsequent additional nucleophilic molecule[36]. Collectively, in contrast to numerous strategies for modifying the N-terminus or side-chain of amino acids, the chemical methods for C-terminal functionalization are scanty and limited. Meanwhile, the well-established C-terminal modification approaches are mainly either enzyme-dependent or only suitable for transpeptidation, which creates a difficult scenario (Supplementary Fig. 1).

The cysteine protease domain (CPD) is one of the core elements of the multifunctional automatic processing repeats in toxin (MARTX) toxin[37–39], which has been widely used as a C-terminal tag to improve the expression, solubility or purification of POIs[40–42]. Compared with other tags applied in protein expression, removing the CPD tag is quite convenient by only a catalytic equivalent of InsP$_6$[41], a commercially available small molecule. As we found, blocking the free cysteine of CPD will inhibit the self-releasing process, indicating that the free cysteine is the key residue for trigging the self-cleavage[43]. Accordingly, we conjectured that a thioester intermediate formed during the process of InsP$_6$-induced CPD cleavage. Thioester, which can further react with hydrazine, amine, cysteine, etc., is an important intermediate in the Native Chemical Ligation (NCL)[44]. Herein, we found that the C-terminus of the POIs could be synchronously modified by an amine-containing molecule during the InsP$_6$-triggered CPD self-cleavage. Examples of specific C-terminal functionalization of POIs with various substrates were also provided, along with their uses in several diverse biological applications, such as nanobody-sugar conjugate, nanobody-drug conjugate, biparatopic antibody, C-terminal fluorescent labeling, etc.

## Results

### Identification of CPD-induced C-terminal functionalization

CPD, essentially, is a cysteine protease that could be activated by the small molecule InsP$_6$. As discussed above, we found that the modification of the sole free cysteine of CPD will inactive its protease activity, demonstrating that the cleavage of peptide bond is Cys dependent[43]. Normally, the mercapto group of cysteine attacks the peptide bond and forms a thioester intermediate which could be further hydrolyzed by water to form a free carboxyl group. In addition, the thioester intermediate is a preferable substrate to react with amine-containing molecules. More recently, Rehm et al. reported the thioester intermediate during [C247A]OaAEP1-catalyzed aminolysis[25]. Hence, we proposed that the InsP$_6$-induced CPD self-cleavage undergoes the thioester intermediate as well, which could be amidated by amines during hydrolysis, and then achieves the C-terminal modification and functionalization of POIs.

To test our hypothesis, we incubated a CPD fusion protein, Endo-F3(D165A)-CPD[40,43] (100 μM), with 3-azidopropylamine (50 mM) in PBS at 4 °C for 30 min with or without (w/o) InsP$_6$ (100 μM) (Fig. 1). The reaction was analyzed by LC-MS to check the molecular formula of Endo-F3(D165A). According to the LC-MS results, incubation of Endo-F3(D165A)-CPD (molecular weight 55616) with InsP$_6$ gave two major LC-MS results (Fig. 1a), 23745 (consistent to CPD) and 31888 (consistent to Endo-F3(D165A)). Meanwhile, incubation of the fusion protein only with 3-azidopropylamine would not result in any molecular weight change of the fusion protein (Fig. 1b), indicating that InsP$_6$ could release CPD from POI but not an amine molecule. However, as shown in Fig. 1c, incubating the fusion protein with 3-azidopropylamine and InsP$_6$ gave a molecular weight of 31970, which equals 31888 (M.W. of Endo-F3(D165A)) + 100 (M.W. of 3-azidopropylamine) − 18 (M.W. of water), and a molecular weight of 23745 (M.W. of CPD). To confirm that the amidation occurs during CPD self-cleavage, we further performed a control experiment in which the fusion protein is first incubated with InsP$_6$ for 2 h and then 3-azidoprpylamine is added. As expected, the LC-MS analysis result (Fig. 1d) is consistent with the one without 3-azidopropylamine (Fig. 1a), demonstrating that the released CPD will not transfer the amine group onto the C-terminus but during the self-cleavage process.

### Condition optimization of the C-terminal modification

Next, we sought to optimize the reaction conditions. First, the model substrate Endo-F3(D165A)-CPD (100 μM) was incubated with 3-azidopropylamine (2–50 mM) in the presence of InsP$_6$ (100 μM) in PBS at 4 °C. The reaction was monitored by LC-MS (Fig. 2) and the

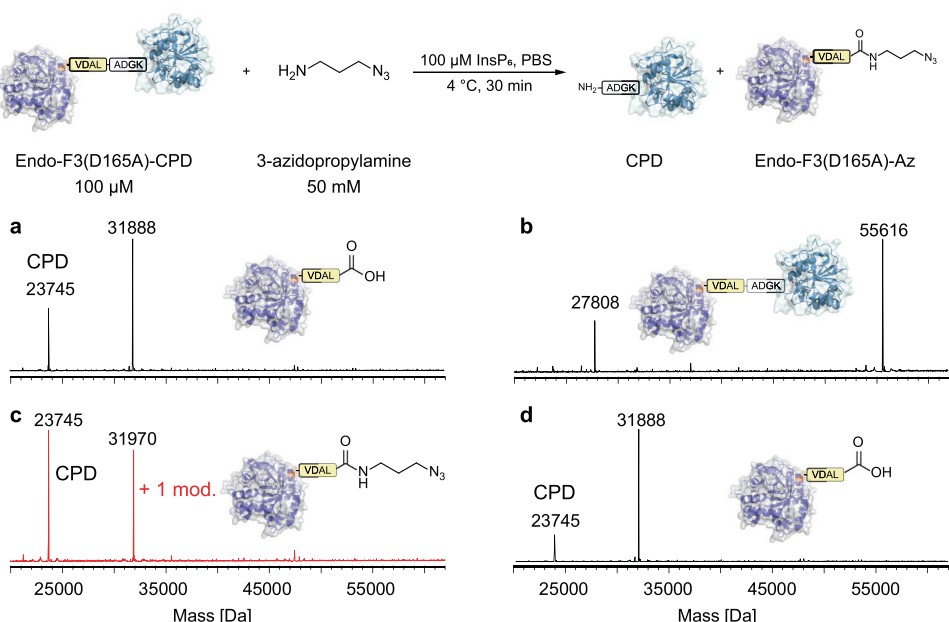

**Fig. 1 | Preliminary evaluation of specific modification of Endo-F3(D165A)-CPD mediated by InsP$_6$.** LC-MS analysis of Endo-F3(D165A)-CPD after incubation with InsP$_6$ (**a**), with 3-azidopropylamine (**b**), with InsP$_6$ and 3-azidopropylamine (**c**) or incubation with InsP$_6$ for 2 h and then 3-azidopropylamine was added (**d**).

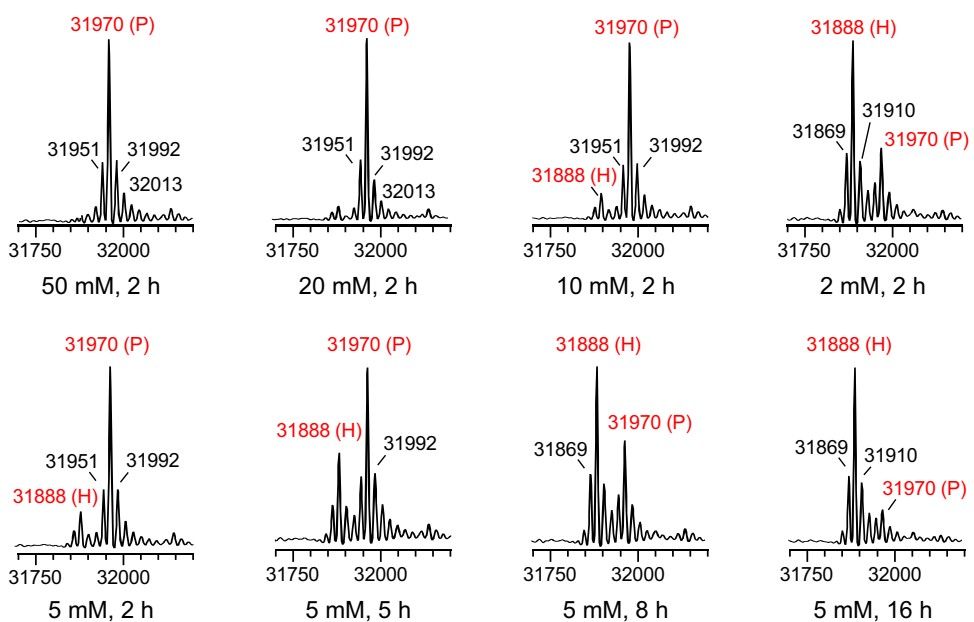

**Fig. 2 | MS spectra profiles of CPD-mediated C-terminal azidation of Endo-F3(D165A) with different concentration of 3-azidepropylamine (100 µM of Endo-F3(D165A)-CPD, x mM of 3-azidopropylamine, 100 µM of InsP$_6$, in PBS at 4 °C).** The reaction was analyzed by ESI-TOF-MS at different time points. P: product; H: hydrolyzed.

percentage of ligation product or hydrolysis product was summarized in Table 1. We found that 20–50 mM of 3-azidopropylamine will complete the reaction within 2 h and give a 100% ligation product (Fig. 2; Table 1, entry 1-2). However, when the amine concentration was lowered to 5 mM, 2 h's incubation gave a yield of 85%, but a 15% hydrolysis product was found (Fig. 2; Table 1, entry 3). Further lowering the amine concentration to 2 mM gave only 30% aminolysis product and 70% hydrolysis product respectively (Fig. 2; Table 1, entry 4). From these results, we could observe that the CPD-induced C-terminal aminolysis is a concentration-dependent reaction, indicating a competitive reaction between the amine and water molecules to the thioester intermediates. Additionally, we found that the percentage of hydrolysate increases in a time-dependent manner (Fig. 2; Supplementary Fig. 2; Table 1, entry 3–5), which suggests a reversible reaction that the C-terminal modification could be again released by CPD and give a free C-terminus of the POI.

InsP$_6$ binds to a conserved basic cleft that is distant from the protease active site, and the binding induces an allosteric switch that leads to the autoprocessing and release of CPD in a concentration-dependent manner[37,38]. Hence, we hypothesized that a lower InsP$_6$-to-substrate ratio may weaken the hydrolysis, and give a better balance between hydrolysis and aminolysis. The evaluation results showed that lowering the InsP$_6$ concentration (500 to 5 µM) will significantly promote the aminolysis yield from 62 to 90% (Fig. 3a, Table 1, entry 4 and 6–10), which is consistent with our speculation. However, both the aminolysis and hydrolysis were inhibited when further decreased InsP$_6$ to 1 µM or 0.2 µM (Fig. 3a, Table 1, entry 11–12). Finally, 5 µM of InsP$_6$ was chosen for further optimization to avoid significant hydrolysis, and the aminolysis product could remain stable in the reaction system for a relatively longer time, which is convenient for subsequent analysis and purification (Supplementary Fig. 3).

Additionally, we performed a pH scan to identify the suitable pH range for CPD-induced C-terminal aminolysis. We chose sodium phosphate buffer (PB) and set the pH at 6.0, 7.0, 8.0, and 9.0 as the final pH value (Supplementary Fig. 4). As shown in Fig. 3b and Table 1 (lane 13–16), we found that the reaction gives 100% hydrolysis when the final pH is <= 7.0, and gives higher aminolysis yield as pH rises. Considering the instability of POIs under strong alkaline conditions, we ultimately chose pH 8.0 as the ideal pH. Further, we also assessed the influence of buffer

systems on the CPD-mediated C-terminal modification. As illustrated in Fig. 3c, despite that the working pH range of NaOAc buffer is 3.6–5.8, the CPD-induced C-terminal modification in NaOAc buffer (pH 8.0, lane 22) performs best, giving the ligation products in 94% yield at 5 h, and 88% at 16 h, an odd phenomenon hard to be explained. However, the employment of HEPES, Tris, MOPs, Tricine, or PB buffer gave a relatively unsatisfied yield (Table 1, entry 17–21, Supplementary Fig. 5). Since some buffer solutions are amine-based, such as Tris, HEPES, and Tricine, we tested if any aminolysis occurred by performing the InsP$_6$-triggered CPD self-cleavage of Endo-F3(D165A)-CPD in these buffers, as a negative control. All of the LC-MS monitoring results are identical and equal to the molecular weight of Endo-F3(D165A), proving that these buffers are compatible with the current reaction (Supplementary Fig. 6).

Subsequently, we investigated if a catalyst can facilitate the CPD-mediated C-terminal modification. MESNA, Thiophenol, 4-mercaptophenylacetic acid (MPAA) that are extensively used in NCLs, as well as N-hydroxybenzotrizole (HOBt), 4-dimethylaminopyridine (DMAP), and 4-pyrrolidinopyridine (PPY) which are frequently used in accelerating acylation reactions, were used for the assessment. First, the ability of these chosen catalysts (5 mM) to accelerate the reaction rate was tested. As indicated in Fig. 4a, a decrease in conversion was observed when MESNA, Thiophenol, or MPAA was used, which is presumably owing to the reversible interaction of these substrates' free thiols with the POI-CPD thioester intermediate. Notably, DMAP and PPY exhibited the ability to promote aminolysis reaction, gave an amidation yield of 76 and 70% respectively, and DMAP showed the best catalytic activity (Supplementary Fig. 7-8). As we know, DMAP and its derivative PPY are strong catalysts for acylation, and our results also indicated that they possess high nucleophilicity towards thioester as well. The formed "acyl-DMAP cation" is a more labile intermediate than thioester to amines, hence improving the reaction rate and yield. Further, we investigated the influence of DMAP concentration. According to the LC-MS results, the aminolysis products predominated when the DMAP concentration rose, while lowering the total reaction efficiency (Fig. 4b, Supplementary Fig. 9). Meanwhile, 5 mM or 10 mM of DMAP showed similar catalytic activity. Hence, 5 mM of DMAP was used, as the catalyst, for further studies. In addition, we also found that CPD-induced C-terminal modification could occur at 4 or 25 °C, indicating a good temperature tolerance (Fig. 4c, Supplementary Fig. 10).

**Table 1 | Optimization of the Reaction Conditions[a]**

| Entry | Endo F3 (D165A)-CPD | 3-azido-propylamine (mM) | InsP$_6$ (µM) | Buffer | 5 h | | 16 h | |
|---|---|---|---|---|---|---|---|---|
| | | | | | Ligation Yield (%) | Hydrolysis Yield (%) | Ligation Yield (%) | Hydrolysis Yield (%) |
| 1 | 100 µM | 50 | 100 | PBS | 100 | 0 | 100 | 0 |
| 2 | 100 µM | 20 | 100 | PBS | 100 | 0 | 100 | 0 |
| 3 | 100 µM | 10 | 100 | PBS | 93 | 7 | 85 | 15 |
| 4 | 100 µM | 5 | 100 | PBS | 67 | 33 | 9 | 91 |
| 5 | 100 µM | 2 | 100 | PBS | 6 | 94 | 0 | 100 |
| 6 | 100 µM | 5 | 500 | PBS | 62 | 38 | 4 | 96 |
| 7 | 100 µM | 5 | 200 | PBS | 68 | 32 | 5 | 95 |
| 8 | 100 µM | 5 | 50 | PBS | 72 | 28 | 7 | 93 |
| 9 | 100 µM | 5 | 20 | PBS | 66 | 34 | 36 | 64 |
| 10 | 100 µM | 5 | 5 | PBS | 90 | 10 | 80 | 20 |
| 11 | 100 µM | 5 | 1 | PBS | 63 | 25 | 59 | 41 |
| 12 | 100 µM | 5 | 0.2 | PBS | 18 | 11 | 37 | 47 |
| 13 | 100 µM | 5 | 5 | PB pH 6.0 | 0 | 100 | 0 | 100 |
| 14 | 100 µM | 5 | 5 | PB pH 7.0 | 0 | 100 | 0 | 100 |
| 15 | 100 µM | 5 | 5 | PB pH 8.0 | 47 | 53 | 49 | 51 |
| 16 | 100 µM | 5 | 5 | PB pH 9.0 | 82 | 18 | 75 | 25 |
| 17 | 100 µM | 5 | 5 | PB | 43 | 57 | 39 | 61 |
| 18 | 100 µM | 5 | 5 | HEPES | 42 | 58 | 38 | 62 |
| 19 | 100 µM | 5 | 5 | Tris | 62 | 38 | 57 | 43 |
| 20 | 100 µM | 5 | 5 | MOPS | 45 | 55 | 40 | 60 |
| 21 | 100 µM | 5 | 5 | Tricine | 45 | 55 | 41 | 59 |
| 22 | 100 µM | 5 | 5 | NaOAc | 94 | 6 | 88 | 12 |
| 23 | 100 µM | 2 | 5 | NaOAc | 12 | 88 | 9 | 91 |

[a] The reactions were performed at 4 °C and corresponding buffer (50 mM, final pH=8.0 unless otherwise indicated). The conversion rates were measured by LC-MS.

## Free CPD could serve as an independent enzyme to mediate C-terminal functionalization and de-functionalization

As discussed above, we observed a reversible reaction that the C-terminal modification will be again released from the product and give a free C-terminus (with carboxyl) when the reaction time is prolonged, indicating that the released CPD is catalytic in the presence of InsP$_6$. To test our hypothesis that free CPD can also act as an independent enzyme to catalyze the amidation, we incubated the isolated aminolysis product, Endo-F3(D165A)-Az, with InsP$_6$ and the purified CPD at room temperature for 12 h. As expected, the LC-MS results (Supplementary Fig. 11a) indicated the release of the azide group from the protein, giving a molecular weight of 31888 that is consistent with the Endo-F3(D165A) with a free C-terminus. However, whether Endo-F3(D165A)-Az was incubated separately with InsP$_6$ or with CPD, the azide group remained on the C-terminus (Supplementary Fig. 11a). Additionally, we blocked the free thiol group of Endo-F3(D165A)-CPD with maleimide-modified peptide to prevent the self-cleavage. And further incubations of the caged POI-CPD were performed individually with InsP$_6$, InsP$_6$ + CPD, or InsP$_6$ + native Endo-F3(D165A)-CPD. According to the LC-MS results, incubation with the InsP$_6$ failed to remove the Cys-blocked CPD from the fusion protein. But, incubation with either InsP$_6$ + CPD or InsP$_6$ + Endo-F3(D165A)-CPD did release the Cys-blocked CPD from the fusion protein, suggesting that the combination of InsP$_6$ and CPD may act as a tool to accomplish the chemoenzymatic protein modification (Supplementary Fig. 11b).

## CPD-induced C-terminal modification is compatible with diverse amine substrates and proteins

With the optimized conditions, we then examined the substrate compatibility of this method with various amine-containing molecules. First, we replaced 3-azidopropylamine with other primary-amine

molecules, such as propargylamine, ethylenediamine, 5-amino-pentan-1-ol, 1-amino-2-methylpropan-2-ol, hydrazide, etc., to react with Endo-F3(D165A)-CPD in the presence of InsP$_6$ and DMAP. The LC-MS analysis suggested a good conversion of the fusion protein to C-terminal functionalized Endo-F3(D165A) with these substrates (Fig. 5a, Supplementary Fig. 12). We also found that PEGylated molecules and labeling tags, biotin for instance, are compatible with the method, but not the secondary-amine molecules (Supplementary Fig. 13a). In addition, the reaction capability was also assessed with different amino acids. However, only glycine can be conjugated onto the C-terminus (Fig. 5a, Supplementary Fig. 13b). Further testing with the dipeptide Gly-Gly or tripeptide Gly-Gly-Gly revealed the successful conjugation of these peptides onto the C-terminus at a favorable yield (Fig. 5a).

Motivated by these results, we turned to fuse the CPD onto more proteins and explore the possibility of CPD-induced C-terminal functionalization. First, diverse POIs with a CPD motif fused to their C-terminus (POI-CPD) were expressed, including a nanobody against Her2 receptor, GFP, Endo-A, Endo-S2, etc. These POIs were successfully constructed and purified from *E.Coli* system (Supplementary Fig. 14). The obtained fusion proteins (100 µM) were incubated with 3-azidopropylamine (5 mM or w/o), InsP$_6$ (5 µM), DMAP (5 mM), respectively. As illustrated in Fig. 5b, c, the fused CPD could be released from these fusion proteins in the presence of InsP$_6$, and the addition of 3-azidopropylamine resulted in the successful and highly efficient C-terminal azidation of the proteins (Fig. 5c, Supplementary Fig. 15-19), demonstrating the universality of this approach.

## Functionalization of target proteins

Compared to the conventional antibody, nanobody possesses unique features, like small molecular weight, high stability, better solubility,

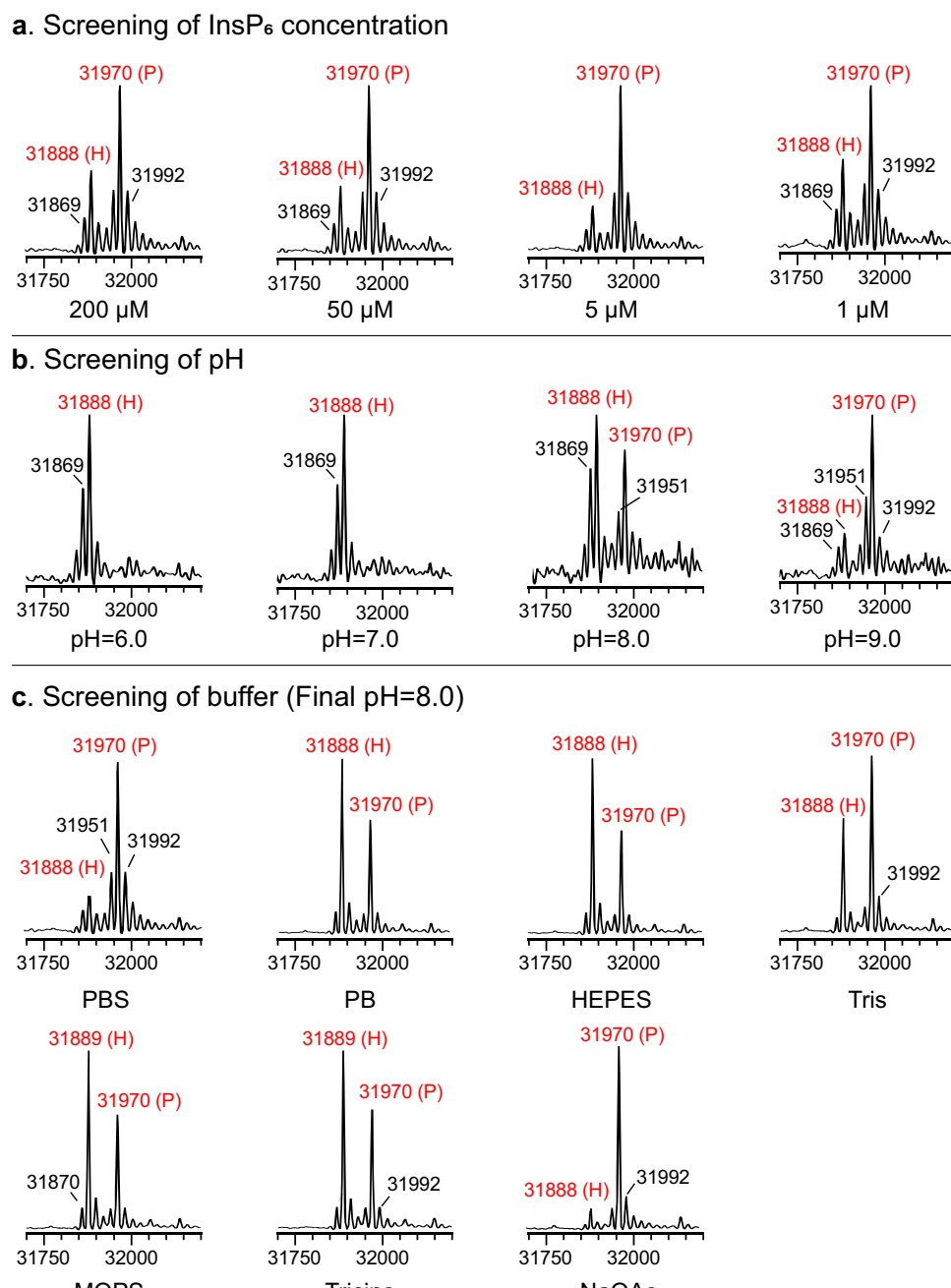

**Fig. 3 | Determination of the CPD-mediated C-terminal azidation of Endo-F3(D165A) under different conditions (100 μM of Endo-F3(D165A)-CPD, 5 mM of 3-azidopropylamine, x μM of InsP₆, buffers, 4 °C, 5 h). a** With different InsP₆ concentration in PBS. **b** In PB with different pH. **c** Under different buffers (final pH = 8.0). P: product; H: hydrolyzed.

and low immunogenicity, which endues nanobody with good tumor penetration ability and easy manufacture. Bioconjugation of a nanobody with functional groups such as fluorophore, peptide, and toxin can improve the efficacy and potency of nanobody and expand its applications in cancer treatment and diagnosis. Therefore, with the established C-terminal functionalization strategy, we next aimed to modify the Her2 nanobody C-terminus for the gain of functions.

In our study, we found that the expression of native Her2 nanobody in *E.Coli* system is scarce. In contrast, the fusion of CPD onto the Nb C-terminus significantly increased the expression yield (from 1-2 mg/L to 200 mg/L). First, under optimal conditions, Her2 Nb-CPD was rapidly hydrazinolyzed by incubating with hydrazine and InsP₆. According to the LC-MS, Nb-hydrazide was quantitively obtained

without hydrolysis (Fig. 6a, Supplementary Fig. 20a). With Her2 Nb-hydrazide in hand, we introduced a GlcNAc onto the C-terminus in an approximately 86% conversion by incubating Nb-hydrazide and GlcNAc in 100 mM NaOAc buffer, pH 4.0 (Fig. 6a, Supplementary Fig. 20b). To test the potential in NCL, we treated the Nb-hydrazide with NaNO₂ and MPAA (mercaptophenylacetic acid) to form the intermediate Nb-MPAA thioester, which then reacted with cysteine and the Flag tag with N-Cys. According to the LC-MS results, the Cys or Cys-containing peptide was successfully ligated onto the C-terminus of Her2 nanobody (Fig. 6b, Supplementary Fig. 20c-d).

At present, Nanobody-Drug Conjugates (NDC) is a booming research field. With Nb-Az in hand, we incubated it with the linker-payload of BCN-Lys(PEG₂₄)-VC-PAB-MMAE[45] (BCN, bicyclo[6.1.0]

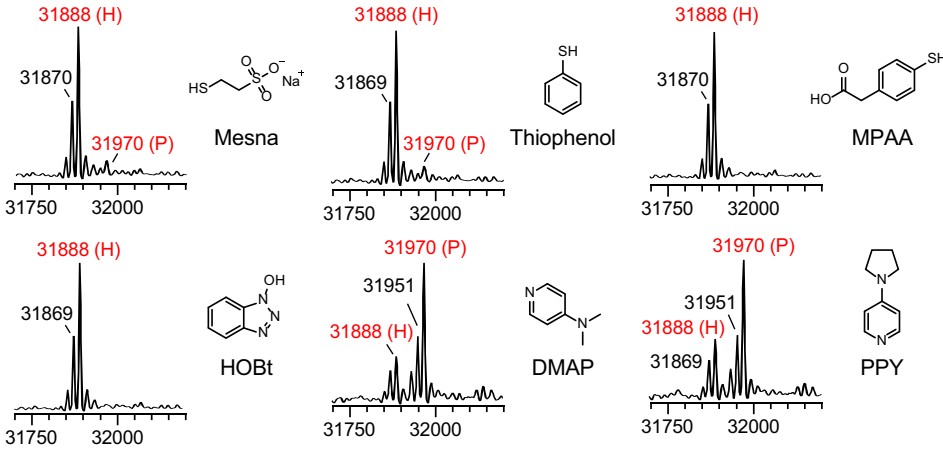

### a. Screening of catalyst

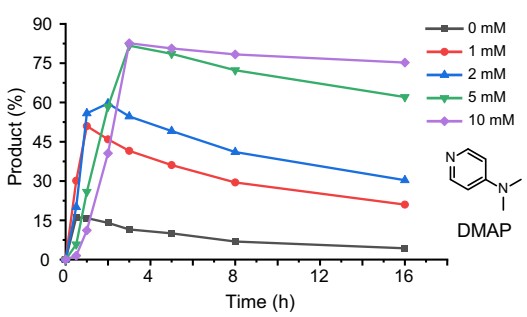

### b. Screening of DMAP concentration

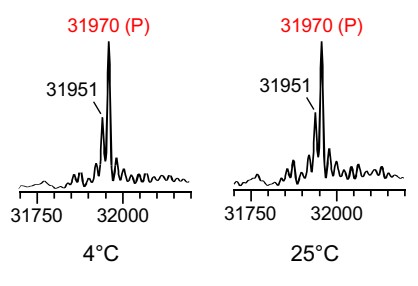

### c. Screening of temperature

**Fig. 4 | Screening of the catalysts for accelerating the CPD-mediated C-terminal functionalization (100 μM of Endo-F3(D165A)-CPD, 2 mM of 3-azidopropyla-mine, 5 μM of InsP6, x mM of catalyst, in NaOAc buffer (final pH=8.0)). a** MS spectra profiles of the CPD-mediated C-terminal azidation of Endo-F3(D165A) with different catalysts (5 mM of catalyst, 4 °C, 3 h). **b** Time course of protein mod-ification with different concentrations of DMAP (0-10 mM, 4 °C, 3 h). **c** MS spectra profiles of the CPD-mediated C-terminal azidation of Endo-F3(D165A) with different temperatures (5 mM of DMAP, 4 °C or 25 °C, 5 h). P: product; H: hydrolyzed.

nonyne; PEG, poly(ethylene glycol); VC-PAB, Valine-Citrulline-para-aminobenzyl alcohol; MMAE: monomethyl auristatin E, a cytotoxin) to afford an NDC that conjugates drug MMAE on the C-terminus (Fig. 7a, Supplementary Fig. 21a-b). In another application, we also synthesized a nanobody-FITC (fluorescein isothiocyanate isomer I, a fluorescence) probe by reacting the Nb-Az with DBCO-FITC (DBCO: dibenzocy-clooctyne) (Fig. 7b, Supplementary Fig. 21c). Since the molecular weight of the Her2 Nb is around 13 kDa, we also used RP-HPLC to identify the modification process. From the HPLC results, we could clearly figure out that either the aminolysis or functionalization step has a good yield (yield > 95%, Fig. 7c). The excess linker-drug/FITC could be simply removed from the reaction mixture by centrifugation, and the pure Her2 NDC or Nb-FITC was efficiently achieved. After that, the in vitro anti-tumor activity of the NDC was evaluated against the Her2-positive cancer cells, SK-Br-3, and Her2-negative cell line MDA-MB-231 as control. As shown in Fig. 7d, e, the NDC, compared to the naked nanobody, exhibited potent and selective anti-tumor activity against the Her2-positive cell line. Meanwhile, the fluorescent SDS-PAGE indicated the successful conjugation of FITC onto the Nb (Fig. 7f). The obtained Nb-FITC was then used in confocal microscopy determination, which could bind to the surface of SK-Br-3 cells but not MDA-MB-231 cells according to the results (Fig. 7g). These findings suggested that the conjugation of functional groups onto the C-terminus will not compromise the target protein's biological func-tions, suggesting the potential for exploiting protein functionalization by the CPD-induced C-terminus modification.

Next, we covalently linked the Her2 Nb onto trastuzumab to construct a biparatopic antibody. Previously, we reported the

traceless site-specific labeling of native antibodies on Fc K248 (EU numbering) using ligand-directed modification[27]. Here, we synthe-sized a DBCO- or MTz-tagged trastuzumab by incubating trastuzu-mab with FcBP-TE-PEG4-DBCO (Supplementary Fig. 23; FcBP, Fc binding peptide; TE, thioester) or FcBP-TE-MTz (Supplementary Fig. 24; MTz, methyl tetrazine) respectively, which (5 mg/mL, 33.4 μM) was further incubated with the Her2 Nb-Az (133 μM, 4 eq) or Nb-TCO (Supplementary Fig. 22, 500 μM, 15 eq; TCO, trans-cyclooctene) at 37 °C for 8 h (Fig. 7h). The reaction mixtures were analyzed by SDS-PAGE. A new band at ~ 70 kDa, which corresponds to the molecular weight of trastuzumab heavy chain plus nanobody, was observed (Fig. 7i, Supplementary Fig. 25), indicating the suc-cessful formation of the biparatopic antibody. According to the results, the conjugation of Trastuzumab with Her2-Nb by TCO-MTz pair is more efficient than the one by DBCO-Az pair, with a yield of 86 *vs* 32% respectively, which provides a feasible approach for chemi-cally constructing bifunctional antibodies.

## Discussion

The chemical C-terminal modification of the protein of interest has long been a challenge because the carboxylic acid group is quite ordinary and is hard to selectively react with the C-terminus. Here, by employing a self-cleavage tag, we developed a facile chemical approach for protein C-terminal modification and functionalization with diverse amine-containing substrates. CPD tag is widely used to increase protein expression. It can be released from the protein in a self-catalytic approach that is triggered by InsP6. We found that during the CPD self-cleaving step, the thioester intermediate will react with the amine

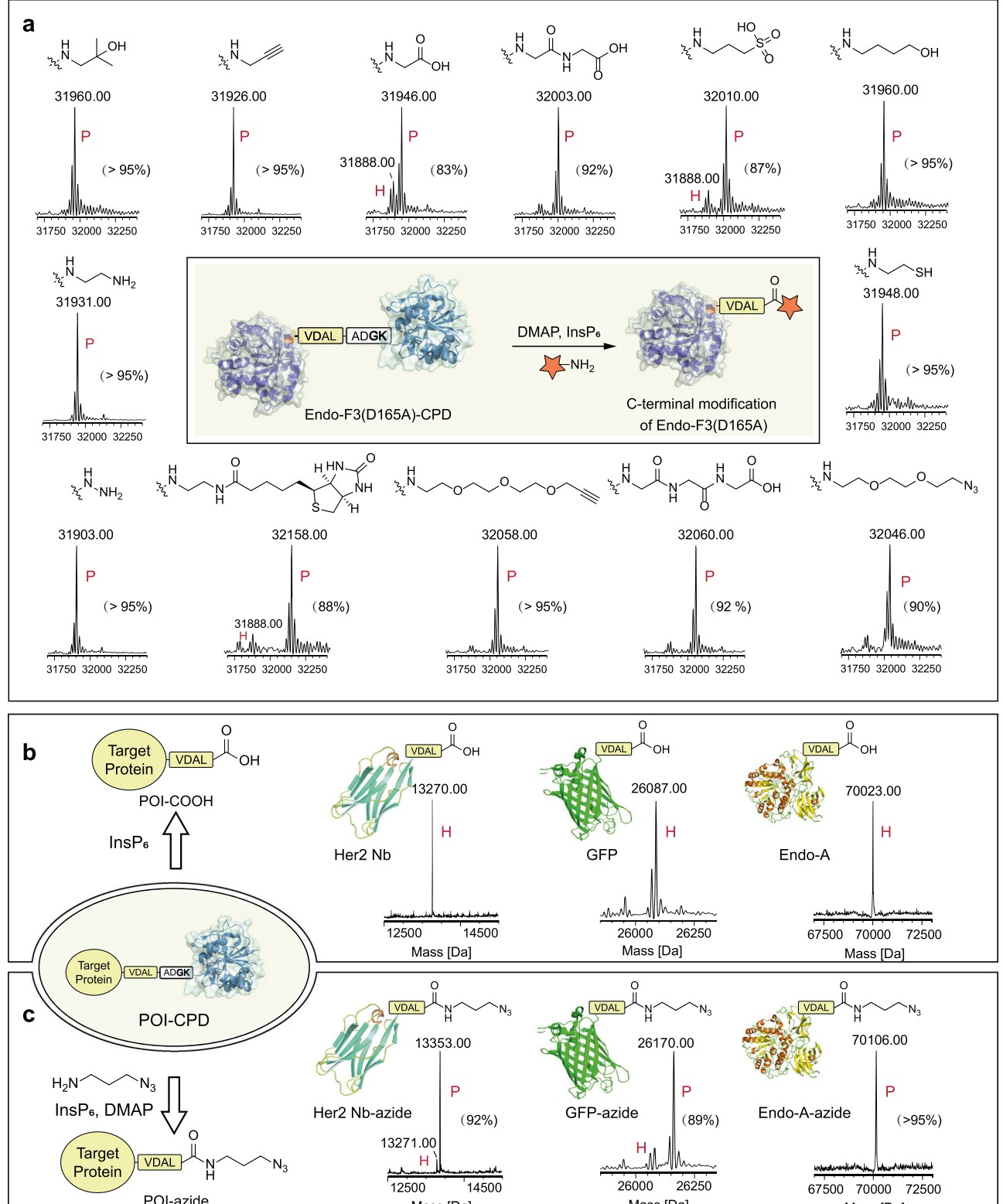

**Fig. 5 | InsP$_6$-mediated ligation to the C-terminus. a** InsP$_6$-mediated ligation of various functional labels to the C-terminus of Endo-F3(D165A). **b** InsP$_6$-mediated hydrolysis of POI-CPD. **c** InsP$_6$-mediated protein labeling of azide group to C-terminus. All modified proteins were analyzed by LC-MS. P: product; H: hydrolyzed.

molecules and form a new amide bond. In this paper, a systemic reaction optimization of the CPD-mediated C-terminal modification was studied, including InsP$_6$ concentration, buffers, pH, and the catalysts. Besides, a reversible reaction was found that the presence of CPD and InsP$_6$ will further hydrolysis the C-terminal modification, enabling a

de-functionalization process. Further, we proved that this approach has a wide compatibility of the amine substrates and can functionalize diverse proteins, such as endoglycosidases, GFP, and the nanobody. By this method, a hydrazide-modified nanobody was efficiently obtained which was ulteriorly functionalized with a GlcNAc or a Flag tag via NCL.

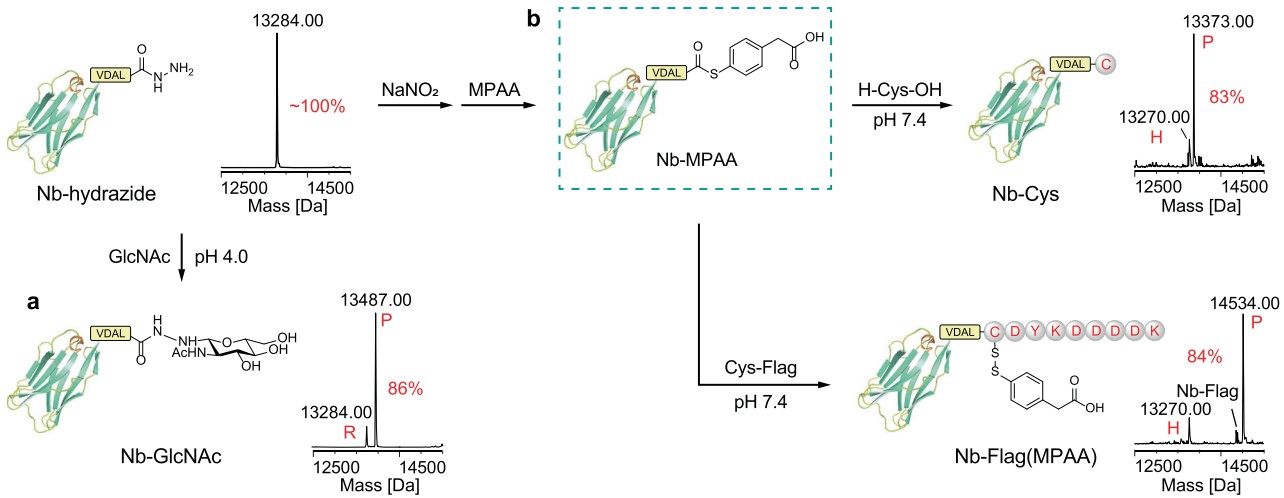

**Fig. 6 | Preparation and characterization of the C-terminal conjugates via Her2 Nb-hydrazide. a** The conjugation of reducing carbohydrate GlcNAc to Her2 Nb-hydrazide. **b** The conjugation of cysteine or peptide to C-terminal by NCL. All modified proteins were analyzed by LC-MS.

Additionally, an azide-tagged nanobody was facilely synthesized via this strategy and was employed to construct nanobody-sugar/drug conjugates, nanobody-fluorescence conjugates, as well as the biparatopic antibody.

As we discussed in the introduction, numerous strategies have been developed to modify the C-terminus of POIs, and lots of them are widely used in the functionalization of target proteins, such as Sortase A, AEPs, etc[46]. However, the existing methods are majorly chemoenzymatic approaches which not only need the expression and purification or isolation of the enzymes but also the design and expression of POIs with fixed C-terminal tags that can be recognized by their unique enzymes. In addition, the donors for modification are majorly amino-acids-based. In other words, these approaches are transpeptidase-catalyzed reactions (Supplementary Fig. 1). Our report, in another aspect, provides a chemical, enzyme-free C-terminal modifying system. In this method, we could achieve C-terminal functionalization by simply incubating POIs-CPD with amines and InsP_6, a chemical trigger, making the modification more convenient and avoiding the expression of additional enzymes. Moreover, the presented method is compatible with amine-containing substrates rather than univocal amino acids (to date, only OaAEP1 was reported that it could serve as a tool to modify the C-terminus with diverse amines[25,26]). Meanwhile, the CPD tag can increase the soluble expression of proteins that are challenging to express, thus providing feasible solutions to broad POIs and applications. For instance, Endo-F3 is a powerful endoglycosidase, but the expression in *E.Coli* system resulted in low yields (<1 mg/L) of soluble enzyme due to the formation of insoluble inclusion bodies, an awkward situation solved by infusing CPD onto the C-terminus (give a yield of > 15 mg/L)[40]. In our report, the CPD tag also raised the expression yield of Her2 nanobody in 100–200 folds (from 1-2 mg/L to 200 mg/L). However, though CPD is currently an effective tag for the expression of non-soluble proteins, the co-expressed CPD (211 amino acids) is a huge domain rather than a small tag, which is a potential problematic for unknown proteins. In addition, the released CPD can continue its activity in hydrolyzing the modified POIs to give a free C-terminus residue, another weakness we need to avoid, a similar phenomenon occurred on PALs but solved by a series of following studies[47–52].

In summary, this approach directly introduces functional tags onto the C-terminus of proteins, enabling rapid product functionalization and diversification in a chemical means. It also implies the possibility of CPD as an efficient protein engineering tool showing a latent application prospect in biotechnology and medicine research, such as nanobody/antibody-drug conjugates.

## Methods
### Materials
The pET26b-SDAB-Her2 nanobody vector and pCMV6-AC-GFP vector was kindly provided by Professor Xiaohua Chen and Associate Professor Sulin Zhang from Shanghai Institute of Materia Medica, respectively. pGEX5-GST, pCW-Endo A, pET26b-Endo-S2, and pET22b-EndoF3(D165A)-CPD vectors were synthesized by GEN-EWIZ (Suzhou, China). All designed primers were synthesized by Sangon Biotech (Shanghai, China). Trans5α and BL21 (DE3) chemical competent cells were purchased from Tsingke (Beijing, China). Other materials used in expression and purification were purchased from Sangon Biotech (Shanghai, China). Sodium phytate (InsP_6), pro-pargylamine, glycyl-glycine, glycyl-glycyl-glycine, biotin, pyrrolidi-nopyridine (PPY), 4-mercaptophenylacetic acid (MPAA), FITC, and 2-(tritylthio)acetic acid were purchased from bidepharm (Shanghai, China). 4-dimethylaminopyridine (DMAP), 3-azidopropylamine, sodium 2-sulfanylethanesulfonate (MESNA), N-hydroxybenzotrizole (HOBt), and 2-aminoethanethiol were purchased from J&K Chemical Ltd (Shanghai, China). Glycine, cysteine, and other amino acids were purchased from GL Biochem (China, Shanghai). Fc binding peptide was purchased from Genscript Biotech Corporation (Nanjing, China). DBCO-CONHS and BCN-O-PNP were purchased from Flechem (Shanghai, China). NH_2-PEG_4-TCO was purchased from Confluore Biotech (Xi'an, China), and MMAE was purchased from Resuperpharmtech (Shanghai, China). Trastuzumab (cat. BDS15-20210301) was ordered and produced by TOT Pharma (Suzhou, China). Centrifugal filtration tube was purchased from Millipore Corporation. Other chemical reagents and solvents, unless indicated, were purchased from Sinopharm Chemical Reagent Co. (Shanghai, China). Nuclear magnetic resonance (NMR) spectra were measured on a Varian-MERCURY Plus-400 or 500 instrument.

### Cell culture
SK-Br-3 (TCHu225) and MDA-MB-231 (TCHu227) cell line was obtained from Cell Bank of Chinese Academy of Sciences (Shanghai, China), which were grown in RPMI-1640 medium supplemented with Penicillin (50 units/mL), Streptomycin (50 μg/mL) and 10% FBS (fetal bovine serum), and were incubated in a water-saturated cell incubator (Thermo Scientific) at 37 °C under 5% CO_2.

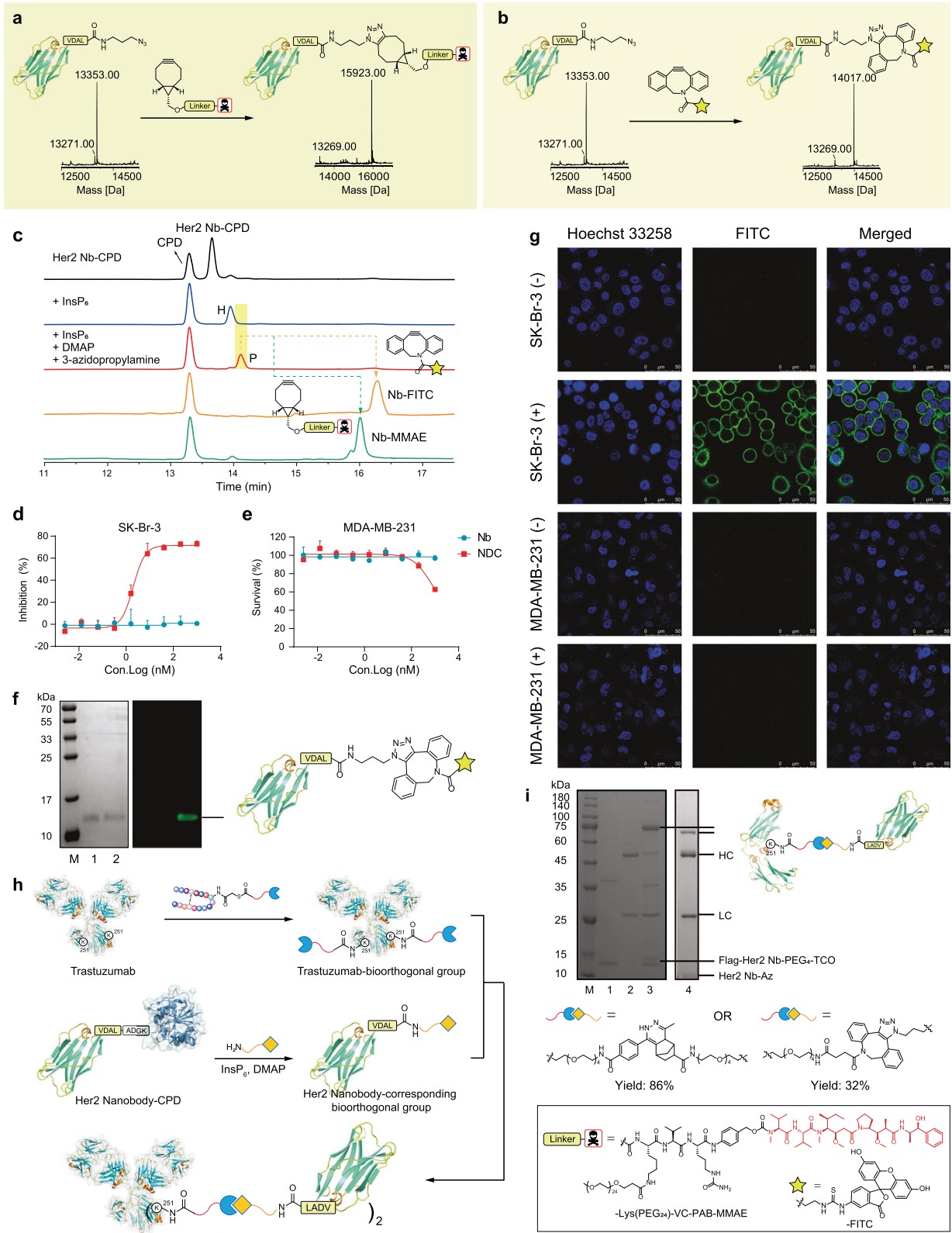

## High-performance liquid chromatography (HPLC)

Analytical RP-HPLC analysis of protein modification was performed on a Thermo Ultimate 3000 instrument with a PIRP-S column (Agilent, 8 μm, 1000 Å) at 70 °C. Mobile phase A: water containing 0.25% TFA; mobile phase B: acetonitrile containing 0.25% TFA. The column was eluted with a linear gradient of 0–100% B in 25 min at a flow rate of 1.0 mL/min.

## Liquid chromatography and electron spray ionization mass spectrometry (LC-ESI-MS)

The LC-ESI-MS data of protein samples were performed and collected on a Waters Xevo G2-XS Q-TOF with a Waters C4 column (ACQUITY UPLC Protein BEH C4, 1.7 μm, 2.1 mm×50 mm) with at 80 °C. Mobile phase A: water containing 0.1% formic acid; mobile phase B: acetonitrile containing 0.1% formic acid; Elution gradient 0-2 min, 5-5% B;

**Fig. 7 | Preparation and characterization of the C-terminal conjugates via Her2 Nb-Az. a** Synthesis of site-specific nanobody drug conjugates with BCN-Lys(PEG24)-VC-PAB-MMAE. **b.** Synthesis of nanobody fluorescent conjugates with DBCO-FITC. **c.** RP-HPLC monitoring of the ligation reaction between Her2 Nb-CPD and 3-azidopropylamine. The ligation reaction was performed at 37 °C for 5 h using 100 μM of Her2 Nanobody-CPD, 5 μM of InsP6, in the absence or presence of 5 mM of DMAP and 5 mM of 3-azidopropylamie. The reaction mixture was purified by centrifugal filtration and then DBCO-FITC or BCN-Lys(PEG24)-VC-PAB-MMAE (5 eq) was added to afford Nb-FITC or Nb-MMAE. P: product; H: hydrolyzed. **d, e** Inhibition of Her2 Nb and Her2 NDC against cell lines SK-Br-3 (**d**, $n = 6000$ cells examined over 3 independent experiments. Data are presented as mean values +/- SEM) and MDA-Mb-231 (**e**, $n = 6000$ cells examined over 3 independent experiments. Data are presented as mean values +/- SEM). **f** Coomassie-stained gel and fluorescence image to show successful conjugation of Nb-FITC. Lane M, molecular weight marker; lane 1, Her2 Nb; lane 2, Her2 Nb-FITC. These experiments are repeated twice independently with similar results. **g** Cell imaging of Her2-positive cancer cell SK-BR-3 and Her2-negative cancer cell MDA-MB-231 stained with Her2 Nb-FITC (nuclei were stained with Hoechst, scale bar 50 μm). These experiments are repeated twice independently with similar results. **h** The scheme of preparation of biparatopic antibodies. **i**. SDS-PAGE analysis of the assembly of Tras-Her2-Nb biparatopic antibodies. Lane M, molecular weight marker; Lane 1, Tras-PEG4-MTz; Lane 2, Her2 Nb-PEG4-TCO, Lane 3, incubation of Tras-MTz and Her2-Nb-TCO; Lane 4, incubation of Tras-DBCO and Her2-Nb-Az. These experiments are repeated twice independently with similar results.

2–10 min, 5–90% B; flow rate: 0.3 mL/min; detecting absorbance: 280 nm.

## SDS-PAGE
Proteins were mixed with 1x protein loading buffer (Reducing or No Deturation & Reducing Buffer, YEASEN) and boiled at 100 °C for 5 min before separation on 12% SDS-PAGE gels. In-gel fluorescence was detected, which was followed by Coomassie staining and imaging on a Bio-Red Chemi Moc MP.

## Cell viability assay
Her2 positive tumor cells SK-Br-3 and Her2 negative tumor cells MDA-MB-231 were seeded in 96-well plates at approximately 6000 cells/well in 90 μL of growth media and incubated overnight at 37 °C in a 5% CO2 cell incubator. Each sample was diluted 5 folds with RPMI 1640 medium from an initial concentration of 0.5 mM, total of 9 concentration gradients were tested. 10 μL of each concentration was added into 3 repeat wells, and the 96-well plates were cultured in a cell incubator with 5% CO2 at 37 °C for three days. Then 10 μL of MTT solution (5.0 mg/mL in PBS) was added and incubated at 37 °C for 4 h, which was followed by 90 μL of 10% SDS solution to dissolve the formazan overnight. Optical density (OD) values were measured at 570 nm using a BioTek Epoch. The $EC_{50}$ values and the cell viability curve were calculated by GraphPad Prism software.

## Cell staining and imaging on a laser scanning confocal microscope
SK-Br-3 cells or MDA-MB-231 cells were seeded in an 8-well chamber with 60,000 cells/well in 200 μL of RPMI-1640 medium, and the plate was incubated at 37 °C with 5% CO2 for 12 h. The cells were washed with wash buffer (PBS containing 5% FBS) 3 times. 200 μL of Her2 Nanobody-FITC (10 μg/mL in wash buffer) was added to the wells and the plate was incubated for another 1 h at 4 °C, then was washed 3 times with wash buffer. The cell nuclei were stained with Hoechst 33258 (Yeasen, 40729ES10, 1: 1000 dilutions, 100 μL) at room temperature for 30 min. All the cells were washed and preserved in the wash buffer on ice at dark. Image analysis was performed on a Leica TCS-SP8 STED instrument. Hoechst 33258 was excited at 405 nm and the nuclei were detected within a window of 415–460 nm. FITC was excited at 488 nm and was detected with a window of 498–551 nm. Blank cells without any pretreatment but stained with Hoechst 33258 were first analyzed while adjusting the HV/gain/offset parameters to give clear nuclei images and an invisible FITC background. Cells from the experimental group was then analyzed with the same parameters as blank cells.

## Homologous recombination of the plasmids expressing various CPD-tagged proteins
A seamless cloning reaction was applied to construct the plasmids expressing CPD-tagged proteins according to the manufacturer's procedures. First, the pET22b plasmid encoding CPD was amplified from pET22b vector encoding EndoF3(D165A)-CPD (forward primer:

5'-GTCGACGCATTAGCGG ATGGAAAAATACTCCATAA-3' and reverse primer: 5'-ATGTATATCTCCTTCTTAAAGTTAA ACAAAATTATTTCTA-GAGG-3'. The resulting PCR products were treated with DpnI enzyme and were purified by gel-extraction kit by the manufacturer's instructions. In addition, distinct linear DNA Inserts encoding different proteins were amplified by PCR from the plasmids of pET26b-SDAB-Her2 nanobody, pGEX5-GST, pCMV6-AC-GFP, pCW-Endo-A, and pET26b-Endo-S2 plasmids respectively with the listed primers. Finally, the constructed linearized CPD-containing pET22b DNA and distinct DNA inserts were mixed respectively (the molar ratio of the DNA insert to linear vector was 2:1) in a seamless cloning master mix, followed by diluting to a specific volume using sterilized ddH2O. The reaction mixture was incubated at 50 °C for 20 min and placed on ice for 2 min immediately.

Forward primer of Her2 nanobody:
5'-GAAGGAGATATACATATGCAAGTTCAGCTGCAGGAAAGCGGT-3'
Reverse primer of Her2 nanobody:
5'-CGCTAATGCGTCGACACCACCGCTGCTCACGGTCACCTGGGT-3'
Forward primer of GST:
5'-GAAGGAGATATACATATGTCCCCTATACTAGGTTATTGGAAAA
TTAAGGG-C-3'
Reverse primer of GST:
5'-CGCTAATGCGTCGACTTTTGGAGGATGGTCGCCACCACCAAA-3'
Forward primer of GFP:
5'-GAAGGAGATATACATATGGAGAGCGACGAGAGCGGC-3'
Reverse primer of GFP:
5'-CGCTAATGCGTCGACTTCTTCACCGGCATCTGCATC-3'
Forward primer of Endo-A:
5'-GAAGGAGATATACATATGTCTACGTACAACGGCCCGCTG-3'
Reverse primer pf Endo-A:
5'-CGCTAATGCGTCGACAAACGAGCCGCTTTTTATGTCGAT-CAT-3'
Forward primer of Endo-S2:
5'-CACCACCACCACCACCACTGAGATCCGGCTGCTAACAAAG-3'
Reverse primer pf Endo-S2:
5'- CCATCCGCTAATGCGTCGACAGCTTTAGCATCATCCATCT-3'
Forward primer of Her2 nanobody (N-flag tag):
5'-GATTACAAGGACGACGATGACAAGCAAGTTCAGCTGCAGGA
AAGCGGTGGT-3'
Reverse primer of Her2 nanobody (N-flag tag):
5'-CTTGTCATCGTCGTCCTTGTAATCCATATGTATATCTCCTTC
TTAAAGTTA-3'

## Expression and purification of the CPD-tagged proteins
The above seamless cloning reaction mixtures were transformed into competent *E. coli* Trans5α respectively. All constructed plasmids were confirmed by DNA sequencing, and either the amino acids sequence or DNA sequence could be found in the Supplementary Information (section 4, protein sequences). The confirmed plasmids with DNA encoding CPD-tagged Endo-F3(D165A), Her2 nanobody, GST, GFP, Endo-A, or Endo-S2 was transformed into the *E. coli* BL21 (DE3) cells for overexpression respectively. The cells were first grown in 5 mL of LB

medium containing 0.1 mg/mL ampicillin at 37 °C for 6 h. When the OD600 of 0.6 was reached, 100 μL of cells were transformed into 100 mL of LB medium containing 0.1 mg/mL ampicillin and the mixture was incubated for another 8 h at 37 °C until the OD600 reached 0.6–1.0, then 0.5 mM of isopropyl β-D-1-thiogalactopyranoside (IPTG) was added to the above cells to induce the expression of the target proteins. The induced cells were incubated at 20 °C for 12 h and harvested by centrifugation at 4 °C and 12000x $g$ for 20 min. The cell lysates were collected via ultrasonication. Distinct CPD-tagged proteins were purified using a Ni-NTA agarose from the supernatant, and the concentration of each protein was measured using a Nanodrop instrument.

### Preliminary evaluation of specific modification of CPD-tagged protein
Endo-F3(D165A)-CPD (100 μM), 3-adizopropylamine (50 mM or without), 100 μM InsP$_6$ was incubated in PBS at 4 °C for 30 min. Or Endo-F3(D165A)-CPD (100 μM) and InsP$_6$ (100 μM) was first incubated in PBS at 4 °C for 2 h, then additional 3-adizopropylamine (50 mM) was added to the reaction mixture which was further incubated at 4 °C for 30 min. All the reactions were monitored by ESI-TOF-MS.

### General procedures for optimized C-terminal modification and purification
CPD-tagged proteins (50–100 μM), amines or hydrazine (5 mM), InsP$_6$ (5 mM), and DMAP (5 mM) were mixed in NaOAc buffer (50 mM, final pH 8.0) at 4 °C for 3 h. The reaction was monitored by ESI-TOF-MS. After completion, the Ni-NTA agarose in PBS was added to remove free CPD with a His6-tag. After agitation for 2 h, the filtrate was collected and concentrated by centrifugal filtration tube (3 kDa or 10 kDa, Millipore). The buffer was exchanged to PBS to facilitate the removal of excess amine, DMAP and InsP$_6$, and to give the C-terminal modification proteins. The ESI-TOF-MS profiles of the C-terminus-modified proteins are listed in Supplementary Table 1 in the supplementary information.

### Modification of Her2 Nanobody with GlcNAc
A solution of Her2 Nb-NHNH$_2$ (50 μM), GlcNAc (1 M) in NaOAc buffer (100 mM, pH 4.0) was incubated at 37 °C for 10 h. ESI-TOF-MS of Her2 Nb-GlcNAc: calcd. 13487.1; found 13487.0.

### Procedures for native chemical ligation
A solution of Her2 Nb-NHNH$_2$ (50 μM), NaNO$_2$ (300 μM) in glycine (0.2 M, pH 3.0) was incubated on ice for 30 min. Then MPAA was added at a final concentration of 20 mM, and the pH was adjusted to 6.0–6.5. The reaction mixture was kept on ice for another 15 min. Then, the cysteine or peptide H-CDYKDDDDK-OH (50 mM) was added and the pH was adjusted to 7.0–7.5 by 500 mM NaOH, and the reaction mixture was incubated at 25 °C for another 2 h. ESI-TOF-MS of Her2 Nb-Cys: calcd. 13373.0; found 13373.0. ESI-TOF-MS of Her2 Nb-Flag: calcd. 14534.2; found 14534.0.

### Preparation of the Her2 Nanobody-FITC conjugate
A solution of Her2 Nb-Az (100 μM), DBCO-FITC (500 μM) in PB buffer (50 mM, pH 7.4) was incubated at 37 °C for 2 h. The reaction was monitored by ESI-TOF-MS. The reaction mixture was purified by centrifugal filtration (10 kDa MWCO, Millipore). The buffer was exchanged to PBS to give the Her2 Nanobody-FITC conjugate in 1.0 mg/mL. ESI-TOF-MS of Her2 Nb-FITC: calcd. 14017.7; found 14017.0.

### Preparation of the Her2 Nanobody-MMAE conjugate
A solution of Her2 Nb-azide (100 μM), BCN-Lys(PEG$_{24}$)-VC-PAB-MMAE (500 μM) in PB buffer (50 mM, pH 7.4) was incubated at 37 °C overnight. The reaction was monitored by ESI-TOF-MS. The reaction mixture was purified by centrifugal filtration (10 kDa MWCO,

Millipore). The buffer was exchanged to PBS to give the Her2 Nanobody-MMAE conjugate. ESI-TOF-MS of Her2 Nb-MMAE: calcd. 15923.1; found 15923.0.

### Preparation of the bispecific antibody
A solution of DBCO-tagged Trastuzumab (5 mg/mL), Her2 Nb-azide (133 μM) in PB buffer (50 mM, pH 7.4) was incubated at 37 °C for 8 h. The progress of the reaction was monitored by SDS-PAGE. In addition, a solution of Mtz-tagged Trastuzumab (5 mg/mL), Her2 Nb-TCO (500 μM) in 50 mM PB buffer (pH 7.4) was incubated at 37 °C for 8 h. The reaction was monitored by SDS-PAGE.

### Reporting summary
Further information on research design is available in the Nature Portfolio Reporting Summary linked to this article.

## Data availability
Methods and all relevant data are available in Supplementary Information. The source data underlying Figs. 4b, c, d, e, f, and i, and Supplementary Figs. S24 are provided as a Source Data file. Source data are provided with this paper.

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

## Acknowledgements

This work was supported by the National Key Research and Development Plan grant (2021YEE0200500), the Natural Science Foundation of China (NSFC, No. 92153301, 82003574, 82204183, 22277126, and 82325045), the Shanghai Sail Program (No. 22YF1457400), the Special Research Assistant Program (Chinese Academy of Sciences, CAS), the Hangzhou innovation and entrepreneurship leading team project (No. TD2020005), the China Postdoctoral Science Foundation (No. 2021M700158), and the Lingang Laboratory (No. LG-QS-202206-03). F. Tang gratefully acknowledge the support of the SANOFI Scholarship Program.

## Author contributions

Y.Z. performed the identification and optimization of CPD-mediated C-terminal azidation, tested different amine substrates and proteins, and constructed the biparatopic antibody. W.S. constructed the fusion proteins of EndoS2-CPD, Endo-A-CPD, GST-CPD, Her2 Nb-CPD, and GFP-CPD, prepared the nanobody-drug conjugates and determined the in vitro activity. Z.L. prepared the nanobody-fluorescence conjugate and

performed the confocal microscopy. Y. W., L.L., J. H., and M.L. gave assistance in expressing POI-CPDs. H.X., and W.Z., helped in the synthesis of chemical substrates and condition optimization. W.H. and F.T. conceived the idea, designed the experiments, supervised the project, analyzed the data, and wrote the manuscript. Y.Z. and W.S. contributed equally to this work.

## Competing interests
The authors declare no competing interests.
