## [Peer Review File · Nature Communications]

C-terminal Modification and Functionalization of Proteins via A Self-cleavage Tag Triggered by A Small MoleculeREVIEWER COMMENTS

Reviewer #1 (Remarks to the Author):

Zeng and colleagues report a new way to modify the C-terminus of proteins of interest. This was achieved by fusing a self-cleavable protease domain to the C-terminus of the protein and inducing cleavage with an allosteric activator, InsP6. The authors demonstrate the usefulness of their method by showing labelling of a number of proteins and by preparing other proteins of potential value, including protein-protein conjugates and antibody drug conjugates. The amount of work presented is quite impressive and on this basis the paper may be suitable for publication in Nature Communications in my opinion.

Conceptually the presented is not particularly novel, the CPD protein domains and their mechanism have been known for some time. The authors exploit the mechanism of the CPD to generate a thioester linked intermediate which can be trapped with small molecule nucleophiles. The concept is very similar to other systems, e.g. inteins and other cysteine proteases (Sortase, AEPs) some of which have been used for more than 20 years. Indeed, despite the authors claim to the contrary, I would argue that C-terminal modification is one of the easiest ways to modify a protein these days with plenty of options out there (see doi: 10.1038/s41570-023-00468-z for a recent review). In the manuscript none of these alternative options are discussed, neither are advantages or disadvantages of the presented method. In addition, the manuscript is full of stylistic and factual errors which make it difficult to follow some of the described experiments.

I have the following suggestions to improve the manuscript:

- 1) The "Results and Discussion" section needs to be expanded. What are the advantages/disadvantages compared to existing methods? The approach requires expression of fusion proteins (unlike other methods (e.g. Sortase), which can be problematic for some proteins? How about the size of the peptide tag that remains in the final product? What's the advantage of having an InsP6 inducible system, if one could simply add a catalyst to the mixture (e.g. Sortase, AEPs)? Many of the acronyms used need to be better explained, e.g. FcBP-TE-PEG4-DBCO and BCN-Lys(PEG24)-VC-PAB-MMAE?
- 2) The authors propose that CPD could be used to de-functionalize protein conjugates but this aspect is not fully explored or discussed. I think this is just a thermodynamically driven side-reaction that from a practical point of view will be difficult to implement because it will require precise kinetic control and optimization of the modification and de-functionalization steps.
- 3) The authors demonstrate aminolysis with a number of amine nucleophiles (Fig 5a), but they ignore the fact that many proteins contain internal Lysine residues which may give rise to lactamization. Indeed some of the mass spec data (e.g. 4a) show prominent MS signals smaller than the hydrolysed product which could be explained by a further dehydration event (lactamization). The mass spec data should be presented in a different way. In most cases (Fig 2,3,4,5) the authors chose large scale overview plots (20 – 60 kDa) to make their case, however, these plots are largely useless to evaluate the finer nuances of the presented reactions (e.g. H, P and other side products). The inset with a much smaller MS range is where the meat is and that's where the focus should be (please label the axis in these inserts). Only the H and P peaks are labelled, but there are quite a few more products that need to be explained or (at the minimum) their masses given.
- 4) All the reactions presented are done in aqueous buffers and will be highly pH dependent. Only in a few cases did the authors specify the pH but it really must be indicated for every experiment shown. Was the pH measured after all components were mixed together?

Reviewer #2 (Remarks to the Author):

In this manuscript, Zeng et al. report a protein C-terminal modification method that is based on the use of the cysteine protease domain (CPD) of *V. cholerae* MARTX toxin. Previous studies have shown that CPD requires InsP6 to be catalytically active and that, when fused to the C-terminal side of a protein of interest, it can catalyze the hydrolysis of a leucyl peptide bond in the same fusion protein, presumably via an acyl-CPD thioester intermediate. The authors of this manuscript find that the thioester intermediate can also be resolved by an amine compound, leading to a C-terminally modified

protein. They show that various unhindered primary amines are effective nucleophile substrates in this CPD-mediated aminolysis reaction. A surprising finding is that a relatively low amine concentration (5 mM) can give an almost exclusive aminolysis reaction with no or little hydrolysis. The method possess some unique features which can make it useful for certain applications, although the requirement for a small amine as the nucleophile also represents a limitation. The work is solid. However, the authors will need to address/answer a number of issues/question before this manuscript can be considered for publication.

The authors characterize this method as a non-enzymatic method, which I disagree. CPD has catalytic activity in the presence of InsP6, just that there is no need for turn-overs as it acts in cis to catalyse the aminolysis reaction. In fact, the authors found that prolonged incubation led to the hydrolysis of the functionalized product, indicating that the released CPD could also act in trans to catalyze the hydrolysis reaction.

It is noticed that all the reactions were done at pH 7.4. Is this the optimal pH? If yes, the authors should provide a pH scan data. Interestingly, the authors also used pH 7.4 in screening different buffers for the reaction, but this pH is beyond the buffering range of some of the buffer systems used. For example, the effective pH range of the acetate buffer is from about 3.7 to 5.7. Outside of this range, the solution would be very sensitive to changes in pH. Considering the small reaction scales, how could the authors ensure that the pH was actually 7.4 for an acetate buffered reaction solution? The authors found acetate to be the best buffer. Is this really due to the nature of acetate ions or to the pH being actually different from 7.4?

Some discussion on why DMAP can increase the aminolysis to hydrolysis ratio needs to be provided. DMAP is known to increase the reactivity of an acylating agent, but by right it also increases the hydrolysis reaction rate. Is it possible that DMAP can bind to CPD to change its catalytic behaviours?

Quantitation of reaction yields and hydrolysis/aminolysis ratios was done using ESI-TOF-MS. While this is acceptable in most cases, it would be necessary to also use an orthogonal quantitation method in at least one experiment. For example, the Her2 nanobody is small enough. An HPLC-based method can certainly be used to determine the aminolysis vs hydrolysis ratio.

What is the affinity of the Her2 nanobody? A nanobody-MMAE conjugate was prepared. However, no information was provided on how the final conjugate was purified. The conjugate exhibited potent cytotoxicity for SK-Br-3 cells (Her2-positive). A control cell line (her2-negative) should also be used for comparison.

An estimated yield of the 8-h click reaction between Tras-PEG4-DBCO and Her2 Nb-Az should be provided based on the SDS gel (Fig 7g).

Protein C-terminal modification can also be done using the intein system. Although the thioester intermediate is not as reactive, intein-mediated hydrazinolysis is reported. Thom, J.; Anderson, D.; McGregor, J.; Cotton, G. Recombinant protein hydrazides: application to site-specific protein PEGylation. *Bioconjug. Chem.* 2011, 22, 1017–1020. Protein C-terminal hydrazinolysis can also be done using a carboxypeptidase. Komiya, C.; Shigenaga, A.; Tsukimoto, J.; Ueda, M.; Mori-saki, T.; Inokuma, T.; Itoh, K.; Otaka, A. Traceless synthesis of protein thioesters using enzyme-mediated hydrazinolysis and subsequent self-editing of the cysteinyl prolyl sequence. *Chem. Commun.* 2019, 55, 7029–7032.

There are numerous grammatical errors throughout the manuscript. Careful proof-reading and editing must be done before resubmission.

Errors with scientific terms:

p3, 3rd line - 'free carbonyl' should be 'free carboxyl'.

Biparatomic or Biaparatomic (Conclusion paragraph)?

Reviewer #3 (Remarks to the Author):

In this work, the team of Zeng et al reported a novel C-terminal modification/functionalisation system. The premise is based on that CPD can mediate InsP6-triggered self-cleavage through a thioester intermediate. Upon the introduction of an amino nucleophile, the POI is labelled at the C-terminus. To prove this concept, the authors did the following experiments:

1. Endo-F3 (D165A)-CPD activity test. Different concentration of nucleophile (lowest substrate:amine ratio was 1:200), buffer, incubation period and small-molecule catalysts tested.
2. There was a section titled "CPD itself could serve as an independent enzyme to mediate C-terminal functionalization and de-functionalization". It describes CPD can mediate the reverse reaction. However, I could understand the latter half of the paragraph due to the grammatical and English mistakes. It was unclear why the abstract indicated that it is non-enzymatic, but they authors also find the hydrolysis of the product is catalytic (mediated by CPD).
3. Different amine and proteins were tested.

I do not endorse publication of this manuscript in Nat Comm or any other similar journal for reasons of two-fold:

1. lack of sound scientific reasoning;
2. lack of high-tier experimental data.

C-terminal protein modification has been well-established in the literature using enzymes and/or chemical reagents (doi.org/10.1039/C8CS00537K; doi.org/10.1021/acs.bioconjchem.2c00411; 10.1021/acs.bioconjchem.1c00442; 10.1039/D0CS01148G; 10.1093/nsr/nwab158). In particular, C-terminal protein and peptide functionalisation by AEP is exceptionally efficient (10.1021/jacs.1c08976). Nearly all of the label tested here have already examined in the indicated literature. There is no clear advantage of the presented system, especially when the substrate:label remains to the low - an issue that has been largely addressed in a recent work (10.1021/jacs.2c13628).

Indeed, many aspects are less ideal than existing protocols. Instead of using enzymes which have superior catalytic efficiency, this work use small molecule as catalysts which no clear advantage. Unlike AEP or sortase, CPD is not catalytic, required to be co-expressed with the gene of interest, and thus the presented labeling system is not atom-economical. In the main text, there is no mentioning about purification, yet the POI must be separated from the reaction which contained a stoichiometric amount of CPD. Unlike AEP (requiring an exposed Asn) and omniligase (requiring a glycy ester), the recognition sequence of CPD is rather long (VDAL). Moreover, the additive InsP6 is not a cheap reagent. AEP and sortase has been used to conduct chemical ligation with fewer trace and chemical steps (10.1039/c5cc07227a). Most importantly, TRACELESS labelling has become standard of protein labelling. The presented system has no clear advantage over existing systems, hence a lack of sound scientific reasoning.

Regarding quality of the data, the issue of hydrolysis is more profound when compared to sortase and AEP. Microscopy of the Her-2 Nb-FITC has low resolution (nucleus is missing? a control with non-specific protein should be included). Yield for most Nb labelling are not included.

Not helping the situation, there were so many grammatical mistakes, making the article difficult to read and review. I suggest the authors subject their articles to proof-reading prior to the next submission. Key references are missing, too. For example, in the main text, it was not clear how the traceless Ig labelling achieved.

In summary, the presented labelling system has significant flaw in design, making it more "awkard" than other C-terminal labelling systems. Unless the authors can explain why a complementary system is needed, otherwise it should not be published in any of the high-tier journals including Bioconjugate Chem, ChemBioChem, Angewandte, JACS, Nat Communication, Nat Chem etc.

Responses to the “REVIEWER COMMENTS”:

Reviewer #1 (Remarks to the Author):

Zeng and colleagues report a new way to modify the C-terminus of proteins of interest. This was achieved by fusing a self-cleavable protease domain to the C-terminus of the protein and inducing cleavage with an allosteric activator, InsP6. The authors demonstrate the usefulness of their method by showing labelling of a number of proteins and by preparing other proteins of potential value, including protein-protein conjugates and antibody drug conjugates. The amount of work presented is quite impressive and on this basis the paper may be suitable for publication in Nature Communications in my opinion.

Answer: *Thanks a lot for the positive comments on our work.*

Conceptually the presented is not particularly novel, the CPD protein domains and their mechanism have been known for some time. The authors exploit the mechanism of the CPD to generate a thioester linked intermediate which can be trapped with small molecule nucleophiles. The concept is very similar to other systems, e.g. inteins and other cysteine proteases (Sortase, AEPs) some of which have been used for more than 20 years. Indeed, despite the authors claim to the contrary, I would argue that C-terminal modification is one of the easiest ways to modify a protein these days with plenty of options out there (see doi: 10.1038/s41570-023-00468-z for a recent review). In the manuscript none of these alternative options are discussed, neither are advantages or disadvantages of the presented method. In addition, the manuscript is full of stylistic and factual errors which make it difficult to follow some of the described experiments.

Answer: *Thanks for corrections. The C-terminal modifications of POIs can be easily achieved in a chemoenzymatic way, and we fully agree the opinions raised by the reviewer. Whereas we want to say that most of the reported C-terminal modification are confined to, at present, the ligation of compounds containing the stated amino acid sequence, e.g. Sortase A ligates the -LPXTG and GGG-, butelase-1 ligates the -NHV and AL-, and OaAEP1 ligates -NGL and GV- (which has been reported recently that the amine-containing small molecules could also be served as the nucleophiles, doi: 10.1021/jacs.1c08976), etc. In the previous manuscript, we tried to tell it is a challenge to modify the C-terminus with non-amino-acid structures. To make it clear, we reorganized this part and discussed the reported C-terminal modification strategies in the revised manuscript.*

I have the following suggestions to improve the manuscript:

1) The “Results and Discussion” section needs to be expanded. What are the advantages/disadvantages compared to existing methods? The approach requires expression of fusion proteins (unlike other methods (e.g. Sortase), which can be

problematic for some proteins? How about the size of the peptide tag that remains in the final product? What's the advantage of having an InsP6 inducible system, if one could simply add a catalyst to the mixture (e.g. Sortase, AEPs)? Many of the acronyms used need to be better explained, e.g. FcBP-TE-PEG4-DBCO and BCN-Lys(PEG24)-VC-PAB-MMAE?

Answer: Thanks for these important suggestions and we revised our manuscript accordingly.

A). as we discussed above, the existing methods for C-terminal modification are majorly chemoenzymatic approaches which need the expression and purification of enzymes, as well as the POIs with fixed amino acid sequences that could be recognized by the corresponding enzymes. In addition, the donors for modification are amino-acids-based. However, **(1)** the method we presented here provides a novel C-terminal modifying system in one hand, in which the “enzyme” CPD is co-expressed with the POIs that will make the process more convenient and get rid of the expression and purification of extra enzymes. Moreover, **(2)** the presented method is compatible with amine-containing substrates rather than univocal amino acid (to date, only OaAEP1 was reported that it could be served as a tool to modify the C-terminus with amines, see 10.1021/jacs.1c08976). Finally, **(3)** CPD can increase the soluble expression of proteins that are challenging to express, thus provides feasible solutions to broad POIs and applications.

B). Considering the disadvantages, as the reviewer said, our approach requires the co-expression of fusion protein CPD (211 amino acids), a potential problematic for unknown proteins. CPD now is an effective tag for the expression of non-soluble proteins, such as the challenging protein EndoF3, nanobody etc. In our report, CPD tag raised the expression yield of Her2 nanobody in 100-200 folds. From this aspect, CPD is not a burdensome tag (in some cases) but a trouble-shooting strategy. In addition, the released CPD can continue its activity in hydrolyzing the modified POIs to give a free C-terminus residue, another disadvantage we need to avoid. Meanwhile, we also found that the released CPD itself could catalyze the capped-CPD from POIs, indicating that CPD could serve as a free enzyme to cleavage certain amino acid sequences which we are studying at present.

We added these discussion of advantage and disadvantages (part A and B) in the “Conclusion” part of revised manuscript.

C). After the self-cleavage of CPD tag, there is 4 amino acids (VDAL) remained on the C-terminus, whose molecular weight is 398 g/mol.

D). InsP6 is a very cheap small molecule ~580\$/100g, while Sortase or other enzymes are proteins that need expression and purification, from which we supposed that the co-expression of CPD and InsP6 inducing system is quite economical and convenient.

E). to make these acronyms clear, we revised the manuscript thoroughly with detailed explanations.

2) The authors propose that CPD could be used to de-functionalize protein conjugates but this aspect is not fully explored or discussed. I think this is just a thermodynamically driven side-reaction that from a practical point of view will be difficult to implement because it will require precise kinetic control and optimization of the modification and de-functionalization steps.

Answer: *We agree with the opinion raised by the reviewer. For the substrate POI-VDALADGK-CPD, the presence of amine substrates and InsP6 results in the products POI-VDAL-R and thermodynamically driven side-product POI-VDAL. However, we could also obtain the product POI-VDAL when we added CPD and InsP6 into the purified product POI-VDAL-R, indicating that POI-VDAL-R could again serve as the substrates of CPD “enzyme”. We revised this part to make it clearly.*

3) The authors demonstrate aminolysis with a number of amine nucleophiles (Fig 5a), but they ignore the fact that many proteins contain internal Lysine residues which may give rise to lactamization. Indeed some of the mass spec data (e.g. 4a) show prominent MS signals smaller than the hydrolysed product which could be explained by a further dehydration event (lactamization). The mass spec data should be presented in a different way. In most cases (Fig 2,3,4,5) the authors chose large scale overview plots (20 – 60 kDa) to make their case, however, these plots are largely useless to evaluate the finer nuances of the presented reactions (e.g. H, P and other side products). The inset with a much smaller MS range is where the meat is and that's where the focus should be (please label the axis in these inserts). Only the H and P peaks are labelled, but there are quite a few more products that need to be explained or (at the minimum) their masses given.

Answer: *Thanks for the nice suggestions. We revised these MS profiles to give a small scale to display the details of peaks. As to the smaller MS signals than the hydrolyzed product, the value difference is exactly 18 and looks like that the dehydration event occurred. However, when we checked the MS profile of native POI-CPD fusion proteins, such as Endo-F3(D165A)-CPD, we could also find that POI-CPD itself contains the smaller molecular weight (-18). Especially for the modified product Endo-F3(D165A)-Az, we could also find the peak with -18. Since the C-terminus has been occupied by the modification, it is impossible for C-terminus to dehydrate again. Meanwhile, the MS data sometimes shows different ratio between the peak abundance of correct product and the product-18. These data and comparison seem to tell that it is MS machine condition dependent and the dehydration more likely occurs within the protein during LC-MS analysis (such as Glu and Lys in the protein sequence). Lastly, we checked other proteins, such as Nb. These data show very clean peak without the peak-18, suggesting that this phenomenon is protein and machine dependent.*

From these LC-MS profiles, we can clearly see the released CPD, POI-Az, and POI itself has the peak-18, and the ratio always changes a little bit between different tests. And it seems that the hydrolyzed product is more likely to give the peak-18 than the Az-product. We can also find that between the two tests of Az-product, the ratio of peak-18 and Az-product peak is different.

4) All the reactions presented are done in aqueous buffers and will be highly pH dependent. Only in a few cases did the authors specify the pH but it really must be indicated for every experiment shown. Was the pH measured after all components were mixed together?

Answer: Thanks for pointing out this issue. To answer this question, we re-conducted these experiments by adding these substrates into 50 mM NaOAc buffer (pH 7.4). The pH will change due to the amine substrate and the final pH is around 8.0. To systemically determine the optimal pH for this process, we did a pH scan using sodium phosphate buffer (6.0, 7.0, 8.0 and 9.0 as the final pH). We found that, as the reviewer said, the reaction is pH dependent and has a best result when the final pH was confined to ≥ 8 , and DMAP indeed increase yield of the aminolysis product. We also revised this part in the new manuscript.

Fig. The LC-MS profile of InsP6-induced CPD cleavage and functionalization with amine of protein EndoF3-D165A. This figure was pasted in the revised manuscript and supporting information (**Fig. 3b**, **Fig. S4**, and **Fig. S7**).

Reviewer #2 (Remarks to the Author):

In this manuscript, Zeng et al. report a protein C-terminal modification method that is based on the use of the cysteine protease domain (CPD) of *V. cholerae* MARTX toxin. Previous studies have shown that CPD requires InsP6 to be catalytically active and that, when fused to the C-terminal side of a protein of interest, it can catalyze the hydrolysis of a leucinyl peptide bond in the same fusion protein, presumably via an acyl-CPD thioester intermediate. The authors of this manuscript find that the thioester intermediate can also be resolved by an amine compound, leading to a C-terminally modified protein. They show that various unhindered primary amines are effective nucleophile substrates in this CPD-mediated aminolysis reaction. A surprising finding is that a relatively low amine concentration (5 mM) can give an almost exclusive aminolysis reaction with no or little hydrolysis. The method possess some unique features which can make it useful for certain applications, although the requirement for a small amine as the nucleophile also represents a limitation. The work is solid. However, the authors will need to address/answer a number of issues/question before this manuscript can be considered for publication.

Answer: *Thanks a lot for the positive comments on our work.*

The authors characterize this method as a non-enzymatic method, which I disagree. CPD has catalytic activity in the presence of InsP6, just that there is no need for turn-overs as it acts in cis to catalyse the aminolysis reaction. In fact, the authors found that prolonged incubation led to the hydrolysis of the functionalized product, indicating that the released CPD could also act in trans to catalyze the hydrolysis reaction.

Answer: *We totally agree with this opinion. CPD is a protease which we can conclude from its name cysteine protease domain. Actually, we are predominately going to regard it as an enzyme-free reaction and the modification could be occur by simply adding a small molecule InsP6, since the “on-position but inactive” enzyme is displayed as an extra “tag” here by fusing it onto the POIs. Honestly, we are investigating the usage of purified CPD and InsP6 as “enzyme” tool to functionalize the POI-VDAL-tag, in which the tag is truncated CPD to make it smaller sequence for CPD cleavage.*

It is noticed that all the reactions were done at pH 7.4. Is this the optimal pH? If yes, the authors should provide a pH scan data. Interestingly, the authors also used pH 7.4 in screening different buffers for the reaction, but this pH is beyond the buffering range of some of the buffer systems used. For example, the effective pH range of the acetate buffer is from about 3.7 to 5.7. Outside of this range, the solution would be very sensitive to changes in pH. Considering the small reaction scales, how could the authors ensure that the pH was actually 7.4 for an acetate buffered reaction solution? The authors found acetate to be the best buffer. Is this really due to the nature of acetate ions or to the pH being actually different from 7.4?

Answer: Thanks for pointing out this issue. Actually, in previous manuscript, the pH 7.4 is the value of solutions without these substrates and the value changed to approximately 8.0 after adding these substrates (mainly caused by the amines). To precisely determined the optimal pH value, we did the pH scan experiment. We used sodium phosphate buffer (the pH range is 5.8-8.0) and set the pH at 6.0, 7.0, 8.0 and a higher one 9.0 (final pH, detected by pH test strips). We found that the InsP6 induced CPD cleavage at all the tested pH conditions, but only hydrolysis occurred when pH ≤ 7.0 . The functionalization occurred when pH > 7.0 and higher pH gave increased yield. Considering the sensitivity of proteins to higher pH (not stable-enough when pH > 8.5), we preferred the optimal range of this reaction within 7.5-8.0.

Fig. The LC-MS profile of InsP6-induced CPD cleavage and functionalization with amine of protein EndoF3-D165A. These figures were pasted in the revised manuscript and supporting information (**Fig. 3b**, **Fig. S4**, and **Fig. S7**).

Some discussion on why DMAP can increase the aminolysis to hydrolysis ratio needs to be provided. DMAP is known to increase the reactivity of an acylating agent, but by right it also increases the hydrolysis reaction rate. Is it possible that DMAP can bind to CPD to change its catalytic behaviours?

Answer: As the reviewer said, DMAP is a good catalyst for acylation reaction. Since DMAP was reported to increase the acylation by forming the intermediate “carboxyl-DMAP cation”, a more sensitive substrate to amines, we supposed that the thiol-ester

intermediate of InsP6-induced CPD cleavage is a potential substrate for DMAP catalyzed acylation as well. According to the results, the addition of DMAP indeed increased the yield and reaction speed of the aminolysis. As we know, amine is a more reactive nucleophile than water. Though the hydrolysis product is a more stable thermodynamic product, the sensitive “carboxyl-DMAP cation” intermediate is more assessable to amines than waters and rapidly gives the aminolysis product, a different situation to that the thiol-ester intermediate is relatively stable and both the aminolysis and hydrolysis reaction is slower. The large amount presence of water molecules become more accessible to the thiol-ester intermediate than the amines, hence the hydrolysis process is more competitive.

Quantitation of reaction yields and hydrolysis/aminolysis ratios was done using ESI-TOF-MS. While this is acceptable in most cases, it would be necessary to also use an orthogonal quantitation method in at least one experiment. For example, the Her2 nanobody is small enough. An HPLC-based method can certainly be used to determine the aminolysis vs hydrolysis ratio.

Answer: Thanks for the great suggestion. Accordingly, we determined the HPLC/HIC characterization of these reactions by RP C18 column or butyl-NPR column. We found that HIC system can monitor the azidation of Endo-F3(D165A), but the baseline and resolution is not friendly to us to use it (see below). The RP-HPLC, however, doesn't give good resolution for large proteins, such as Endo-F3/A/S2 (data not shown) and GFP, but indeed gives a good separation of Nb and Nb-Az. So we majorly added the monitoring of Nb reactions by RP-HPLC in the revised manuscript.

HIC analysis of C-terminal modification of Endo-F3(D165A)

RP-HPLC analysis of C-terminal modification of Nb

RP-HPLC analysis of C-terminal modification of GFP

What is the affinity of the Her2 nanobody? A nanobody-MMAE conjugate was prepared. However, no information was provided on how the final conjugate was purified. The conjugate exhibited potent cytotoxicity for SK-Br-3 cells (Her2-positive). A control cell line (her2-negative) should also be used for comparison.

Answer: 1), The tested Her2 nanobody is 2Rs15d, which was reported with an affinity of $K_d = 3.99 \pm 0.04$ nmol/L to Her2 receptor by SPR (ref: 10.1158/1078-0432.CCR-17-0310), and we added this reference in the revised manuscript. **2),** Meanwhile, we checked the experimental section and added experimental details, including the issue raised by the reviewer. **3),** We compared the activity of Nb-MMAE conjugate against Her2 positive cell line SK-Br-3 and Her2 negative cell line MDA-MB-231 in the revised manuscript.

An estimated yield of the 8-h click reaction between Tras-PEG4-DBCO and Her2 Nb-Az should be provided based on the SDS gel (Fig 7g).

Answer: Thanks for the great suggestion. We revised the figure in the new manuscript.

Protein C-terminal modification can also be done using the intein system. Although the

thioester intermediate is not as reactive, intein-mediated hydrazinolysis is reported. Thom, J.; Anderson, D.; McGregor, J.; Cotton, G. Recombinant protein hydrazides: application to site-specific protein PEGylation. *Bioconjug. Chem.* 2011, 22, 1017–1020. Protein C-terminal hydrazinolysis can also be done using a carboxypeptidase. Komiya, C.; Shigenaga, A.; Tsukimoto, J.; Ueda, M.; Mori-saki, T.; Inokuma, T.; Itoh, K.; Otaka, A. Traceless synthesis of protein thioesters using enzyme-mediated hydrazinolysis and subsequent self-editing of the cysteinyl prolyl sequence. *Chem. Commun.* 2019, 55, 7029–7032.

Answer: *Thanks for the great suggestion. We discussed these works and added the references in the new manuscript.*

There are numerous grammatical errors throughout the manuscript. Careful proof-reading and editing must be done before resubmission.

Errors with scientific terms:

p3, 3rd line - ‘free carbonyl’ should be ‘free carboxyl’.

Biparatopic or Biaparatomic (Conclusion paragraph)?

Answer: *Thanks for the great suggestion. We revised these issues (change “carbonyl” to “carboxyl”, and corrected “biparatopic”) in the new manuscript. Since we are not native English speaker, we are going to seek help from the editorial board to make our manuscript readable.*

Reviewer #3 (Remarks to the Author):

Answer: *Thanks a lot for the review and opinions on our work. We try to answer the comments and questions one by one.*

In this work, the team of Zeng et al reported a novel C-terminal modification/functionalization system. The premise is based on that CPD can mediate InsP6-triggered self-cleavage through a thioester intermediate. Upon the introduction of an amino nucleophile, the POI is labelled at the C-terminus. To prove this concept, the authors did the following experiments:

1. Endo-F3 (D165A)-CPD activity test. Different concentration of nucleophile (lowest substrate: amine ratio was 1:200), buffer, incubation period and small-molecule catalysts tested.
2. There was a section titled "CPD itself could serve as an independent enzyme to mediate C-terminal functionalization and de-functionalization". It describes CPD can mediate the reverse reaction. However, I could understand the latter half of the paragraph due to the grammatical and English mistakes. It was unclear why the abstract indicated that it is non-enzymatic, but they authors also find the hydrolysis of the product is catalytic (mediated by CPD).

Answer: *Sorry for the misunderstanding owing to my bad organization and English writing level. In a major part of our work, we tried to explain that we could add functional groups onto the C-terminus of a POI simply by InsP6 triggered CPD self-cleavage. Since the "inactive" CPD was co-expressed with POIs, the modification of POIs was simply achieved by incubating POI-CPD with amines and InsP6, which means that we don't need to add extra enzyme into the reaction mixture. Hence, we regarded this approach as non-enzymatic. This is the first time to report that CPD could serve as a tag for C-terminus modification and is different from other well-known C-terminus modification strategies which are predominantly enzyme-required and need the expression of POIs with certain tags. So, in the abstract, we emphasized the enzyme-free characterization. However, during our research, we found that the released CPD could further hydrolyze the product in the presence of InsP6, indicating that the released CPD (in the presence of InsP6) could serve as an independent enzyme. That means, InsP6 changes the CPD structure and turns its feature from inactive to active "enzyme". We revised our manuscript to make it readable as much as possible.*

*In addition, we, initially, indeed need **200 times-fold** of amine over POI substrate to give a good yield over 95%. But after the condition optimization, though still needs to be improved, we could achieve similar yield with **20 times-fold** of amine (10 times-fold increase compared to the initial conditions), with DMAP as catalyst.*

3. Different amine and proteins were tested.

I do not endorse publication of this manuscript in Nat Comm or any other similar journals for reasons of two-fold:

1. lack of sound scientific reasoning;

2. lack of high-tier experimental data.

C-terminal protein modification has been well-established in the literature using enzymes and/or chemical reagents (doi.org/10.1039/C8CS00537K; doi.org/10.1021/acs.bioconjchem.2c00411; 10.1021/acs.bioconjchem.1c00442; 10.1039/D0CS01148G; 10.1093/nsr/nwab158). In particular, C-terminal protein and peptide functionalization by AEP is exceptionally efficient (10.1021/jacs.1c08976). Nearly all of the label tested here have already examined in the indicated literature. There is no clear advantage of the presented system, especially when the substrate: label remains to the low - an issue that has been largely addressed in a recent work (10.1021/jacs.2c13628).

Indeed, many aspects are less ideal than existing protocols. Instead of using enzymes which have superior catalytic efficiency, this work uses small molecule as catalysts which no clear advantage. Unlike AEP or sortase, CPD is not catalytic, required to be co-expressed with the gene of interest, and thus the presented labeling system is not atom-economical. In the main text, there is no mentioning about purification, yet the POI must be separated from the reaction which contained a stoichiometric amount of CPD. Unlike AEP (requiring an exposed Asn) and omniligase (requiring a glyceryl ester), the recognition sequence of CPD is rather long (VDAL). Moreover, the additive InsP6 is not a cheap reagent. AEP and sortase has been used to conduct chemical ligation with fewer trace and chemical steps (10.1039/c5cc07227a). Most importantly, TRACELESS labelling has become standard of protein labelling. The presented system has no clear advantage over existing systems, hence a lack of sound scientific reasoning.

Answer: *It's our regret to receive this opinion.*

*1), At present, as summarized as the reviewer, C-terminus modification strategy has been well established. However, it is still full of challenge, since the reported C-terminal labelling systems, always, are enzyme-dependent. When we are going to install target modifications onto the C-terminus, if use these systems, we **not only need to express and purify the POIs with certain tags as required, but also express, purify, or extract the foremost tool, enzyme, such as Sortase A, Butelase-1, OaAEP1, etc.** To some extent, these strategies are not atom-economical as well (need to express both the POI-tag and the enzyme). So, to solve this problem, we attempt to establish a system that can achieve C-terminus modification simply by adding chemical reagents. In our study, we only need to express and purify the POIs with CPD tag, an inactive "enzyme" whose activity can be recovered by a cheap chemical reagent InsP6. **Though some conditions need to be optimized to promote the reaction, this is the first report, to our best knowledge, that C-terminus modification is achieved by simply incubating POIs with small molecules, which we supposed is a well complementary system for chemo-enzymatic C-terminus modification.** Meanwhile, lots of POIs have poor expression yield in E.Coli system, the fusion of C-terminus CPD has been reported as an efficient means to solve this problem, which guarantees the sufficient investigation of POIs C-terminal modification and their further applications (such as Her2 Nb has a yield of 1-2mg/L, while the Her2 Nb-CPD has a significant enhancement to 200 mg/L). Finally, as we discussed in the manuscript, CPD itself could also act as an independent enzyme to*

hydrolyze the POI-tag or the C-terminal functionalized POI (also need the addition of InsP6 to trigger the reaction), and we are studying on the tag size to supply another complementary system to make it atom-economical.

2), Since CPD has a His6 tag, the POIs with C-terminal modification could be achieved by Ni-NTA purification and these small molecules were removed during concentration.

3), InsP6 is a very cheap small molecule ~580\$/100g and the usage in our research is quantitatively less enough, while Sortase or other enzymes are proteins that need expression and purification (a higher price if purchase), from which we supposed that the co-expression of CPD and InsP6 inducing system is quite economical and convenient.

4), Yes, as we know, AEPs have been well studied and lots of optimization and promotion have been reported. However, as a new finding that CPD could also act as a tool for C-terminal modification, this strategy, we believe, will become more efficient and powerful in the future as more studies were performed.

Regarding quality of the data, the issue of hydrolysis is more profound when compared to sortase and AEP. Microscopy of the Her-2 Nb-FITC has low resolution (nucleus is missing? a control with non-specific protein should be included). Yield for most Nb labelling are not included.

Answer: Thank you for your comments. 1), Actually, in our optimal conditions, the hydrolysis is under control and gives the C-terminus modified product in an inspiring yield (more than 95% according to the LC-MS and HPLC). 2), We again did the confocal microscopy of Her2 Nb-FITC against Her2 positive cell line SK-Br-3 and revised these issues. To prove the selectivity, we also performed the microscopy of Her2 Nb-FITC against Her2 negative cell line MDA-MB-231 and found no fluorescence (see Fig. 7). 3), We also added the yield for protein modifications in the revised manuscript.

Hydrophobic interaction chromatography (HIC) analysis also indicated the high yield of CPD mediated C-terminus modification

RP-HPLC analysis of C-terminal modification of Nb reveals excellent yield for azidation of Nb and the successive functionalization.

RP-HPLC analysis of C-terminal modification of GFP reveals good yield for azidation of GFP and the successive functionalization.

Not helping the situation, there were so many grammatical mistakes, making the article difficult to read and review. I suggest the authors subject their articles to proof-reading prior to the next submission. Key references are missing, too. For example, in the main text, it was not clear how the traceless Ig labelling achieved.

Answer: *Thanks for these suggestions. So sorry for these mistakes to make our paper obscure. Since we are not native English speaker, we are going to seek help from the editorial board to make our manuscript readable. Meanwhile, we added more experimental details in the revised manuscript to clear our findings and experimental design and procedures.*

In summary, the presented labelling system has significant flaw in design, making it more "awkard" than other C-terminal labelling systems. Unless the authors can explain why a complementary system is needed, otherwise it should not be published in any of the high-tier journals including Bioconjugate Chem, ChemBioChem, Angewandte, JACS, Nat Communication, Nat Chem etc.

Answer: *As we discussed above, C-terminus modification is still full of challenge, since the reported C-terminal labelling systems always are enzyme-dependent. When using these systems, we **not only need to express and purify the POIs with certain tags, but also the foremost tool enzyme.** To some extent, these strategies are not atom-economical as well. In our study, we only need to express and purify the POIs with CPD tag, which can be triggered by a cheap chemical reagent InsP6. **Though some conditions need to be optimized to promote the reaction, this is the first report, to our best knowledge, that C-terminus modification is achieved by simply incubating POIs with small molecules, which we supposed is a well complementary system for chemo-enzymatic C-terminus modification.** Meanwhile, the fusion of C-terminus CPD has been reported as an efficient means to solve the insufficient expression level of some POIs, which guarantees the sufficient investigation of POIs C-terminal modification and their further applications.*

REVIEWER COMMENTS

Reviewer #1 (Remarks to the Author):

The authors have made some improvements to their manuscript, but more work is needed. The draft is still full of stylistic and grammatical errors and needs to be professionally edited. Some suggestions can be found below (by no means complete). The described system is highly similar to intein fusion proteins that also can excise themselves out of a POI and have been developed by (primarily by Tom Muir's group) to allow for facile N- and/or C-terminal modification (and they have been around for more than 25 years). Unfortunately, these inteins do not get much credit in the current version.

Line 42: "Graham Cotton et al employed..." change to Thom et al. employed

Line 44: "developed by Wenshe Ray Liu et al" change to Qiao et al.

Line 53: "which has been widely used as a C-terminal tag" needs references

Line 77: "Thomas Durek et al" change to "Rehm et al." (notice punctuation of et al. and change throughout).

Line 79: amidation instead of amination

Line 88: To test our hypothesis

Line 89: specify the exact reaction conditions, buffer, pH, temperature, time.

Line 136: Further, because of the co-catalytic effector of InsP6, we hypothesized that a lower InsP6-to-substrate ratio may weaken the hydrolysis, a "side-reaction".

Unclear and needs further explanation. What exactly is the role of the InsP6? Is it really co-catalytic? And how exactly will InsP6 affect hydrolysis?

Figure 1: In the scheme, InsP6 is shown as a reactant, but it is unclear what happens to it? Change it to the format shown in Fig 2.

The reaction scheme shown on top of Fig 1,2,3 and 4 is largely the same and highly redundant. Probably one is enough?

Table 1: Some of these buffers (namely TRIS, HEPES, TRICINE) contain amines and are used in substantial excess relative to the intended nucleophile. Comment, in the paper, on whether these amines can aminolyse the thioester under these conditions. The MS data should tell.

Line 178: "Notably, both DMAP and PPY exhibited an increased aminolysis efficiency, 76% and 70% respectively" What values are these numbers compared to?

Line 195: "To testify our hypothesis..." What is the hypothesis? Specify! Also change testify to test.

Line 222: "and big molecule, Biotin for instance" rephrase, biotin isn't much larger than some of the other molecules shown.

Line 254: Explain what MPAA stands for. If it stands for mercaptophenylacetic acid, then the structures in 6b are wrong.

Figure 6: give the measured, precise pH used in the reaction, not a pH range (7.0-7.5).

Line 264-286: This paragraph doesn't really add much to the present story. The SPAAC chemistry of azide modified proteins is fairly standard. Hence, this paragraph could and probably should be shortened.

Reviewer #2 (Remarks to the Author):

Zeng et al. addressed all the points/concerns raised by the reviewers in the revised manuscript and rebuttals. The revision has helped improve the quality of their work significantly. For example, they conducted the pH scan experiment as I suggested, and the results cleared the doubts on the optimal

pH of the concerned reaction. I don't see any major issues in the revised manuscript and would recommend publication after a few minor points are taken care of.

1) Graphic abstract: One of the R groups is listed as hydrazone, but it is actually a hydrazide.

2) Legend of Figure 3: description for (c) is missing.

3) Table 1: On the Buffer column, 'pH' should be added before the numeric value in PB 6.0, PB 7.0, PB 8.0, PB 9.0. For example, PB pH 6.0.

The footnote of Table 1: aThe reactions were performed at 4 °C and corresponding buffer (50 mM, final pH=8.0 unless otherwise indicated).

4) There are still many grammatical errors in the manuscript. To give just one instance: Line 30, 1st sentence: "Chemically modification ..." should be "Chemical modification ..."

I trust the editorial office will help the authors to fix the English problems.

Signed by Chuan-Fa Liu

Responses to the “REVIEWER COMMENTS”:

Reviewer #1 (Remarks to the Author):

The authors have made some improvements to their manuscript, but more work is needed. The draft is still full of stylistic and grammatical errors and needs to be professionally edited. Some suggestions can be found below (by no means complete). The described system is highly similar to intein fusion proteins that also can excise themselves out of a POI and have been developed by (primarily by Tom Muir's group) to allow for facile N- and/or C-terminal modification (and they have been around for more than 25 years). Unfortunately, these inteins do not get much credit in the current version.

Answer: *Thanks for the review and comments. First, we again tried our best to revise the stylistic and grammatical errors in the second revised version, and we will ask the editorial office for additional help if it is accepted. Second, we sincerely thank the kind help from the reviewer to give some corrections of our manuscript. **Meanwhile, the comment about intein system is quite useful to improve our manuscript. We again revised this part in the 2nd revision, adding “Split intein system has been well-developed in the past two decades, especially by Tom W. Muir; to achieve the semi-synthesis or N/C-terminal modification of proteins, and becomes an important chemical tool to study protein functions. The intein domain can be self-split or removed by MESNA (sodium 2-sulfanylethanesulfonate) and gave a thioester intermediate which could further react with a N-Cys-containing protein/peptide or small molecule.” and “Meng-Jung Chiang et al also used intein system to construct the Fc Domain protein–small molecule conjugate by fusing intein onto the C-terminus of Fc which was further converted into Fc-thioester by MESNA and react with N-Cys-linker-small molecule.”***

Line 42:” Graham Cotton et al employed...” change to Thom et al. employed

Line 44 :” developed by Wenshe Ray Liu et al” change to Qiao et al.

Answer: *Thanks for the corrections, and we already revised these issues in the 2nd revision.*

Line 53: “which has been widely used as a C-terminal tag” needs references

Answer: *Thanks for the corrections, and we added 3 references that use CPD to enhance the expression of target proteins (J Biol Chem 291, 9356-9370 (2016); PLoS One 4, e8119 (2009); BMC Biotechnol 17, 1 (2017).) in the 2nd revision.*

Line 77: “Thomas Durek et al” change to “Rehm et al.” (notice punctuation of et al. and change throughout).

Line 79: amidation instead of amination

Line 88: To test our hypothesis

Line 89: specify the exact reaction conditions, buffer, pH, temperature, time.

Answer: *Thanks for the corrections, and we already revised these issues in the 2nd revision.*

Line 136: Further, because of the co-catalytic effector of InsP6, we hypothesized that a lower InsP6-to-substrate ratio may weaken the hydrolysis, a “side-reaction”. Unclear and needs further explanation. What exactly is the role of the InsP6? Is it really co-catalytic? And how exactly will InsP6 affect hydrolysis?

Answer: *We revised our description of this part. According to the report (**Science** 2008, 322, 265-268 and **J Biol Chem** 2008, 283, 23656-23664), InsP6 binds to the CPD and induces the allosteric switch that leads to the autoprocessing and release of toxin-effector domains. Meanwhile, the release was found in a InsP6 concentration-dependent manner that lower InsP6 concentration decreased the reaction rate and cleavage yield. Hence, we thought that lowering the reaction rate may contribute to the aminolysis process since the amines are competed with waters. We revised this part into “InsP6 binds to a conserved basic cleft that is distant from the protease active site, and the binding induces an allosteric switch that leads to the autoprocessing and release of CPD in a concentration-dependent manner. Hence, we hypothesized that a lower InsP6-to-substrate ratio may weaken the hydrolysis, a “side-reaction”, and give a better balance between hydrolysis and aminolysis.”*

Figure 1: In the scheme, InsP6 is shown as a reactant, but it is unclear what happens to it? Change it to the format shown in Fig 2.

The reaction scheme shown on top of Fig 1,2,3 and 4 is largely the same and highly redundant. Probably one is enough?

Answer: *Thanks for the corrections. We revised the Fig.1 to move InsP6 above the arrow and removed the reaction schemes in Fig 2,3,4 in the 2nd revision.*

Table 1: Some of these buffers (namely TRIS, HEPES, TRICINE) contain amines and are used in substantial excess relative to the intended nucleophile. Comment, in the paper, on whether these amines can aminolyse the thioester under these conditions. The MS data should tell.

Answer: *Yes, it is a good question. To prove that these amine-containing buffers can't aminolyse the thioester, we simply incubated the POI-CPD with InsP6 in these buffers and the LC-MS gave the same value that equals to POI-COOH but not the one with aminolysis. We also added the sentence “Since that some buffer solutions are amine-based, such as Tris, HEPES, Tricine, we conducted the InsP6 triggered CPD self-cleavage of EndoF3(D165A)-CPD in these buffers, as negative control, to test if any aminolysis occurred. The LC-MS monitoring of all these reactions gave the same results*

which are equal to the molecular weight of EndoF3(D165A), indicating that these buffers are compatible to current reaction (Fig. S6).” in the 2nd revision.

Line 178: “Notably, both DMAP and PPY exhibited an increased amidolysis efficiency, 76% and 70% respectively” What values are these numbers compared to ?

Answer: *Actually, the number 76% or 70% is the yield of amidation reaction, and we revised the context in the new manuscript to “both DMAP and PPY exhibited an increased amidolysis efficiency, gave amidation yield of 76% and 70% respectively,”*

Line 195: “To testify our hypothesis...” What is the hypothesis? Specify! Also change testify to test.

Answer: *Thanks for suggestion. We revised this sentence to “To test our hypothesis that free CPD could also serve as an independent enzyme to mediate the amidation, we incubated the isolated...”*

Line 222: “and big molecule, Biotin for instance” rephrase, biotin isn’t much larger than some of the other molecules shown.

Answer: *Thanks for pointing out this issue. We revised the sentence to “with PEGylated molecules and labeling tags, Biotin for instance,”*

Line 254: Explain what MPAA stands for. If it stands for mercaptophenylacetic acid, then the structures in 6b are wrong.

Figure 6: give the measured, precise pH used in the reaction, not a pH range (7.0-7.5).

Answer: *Thanks for these corrections, and we already revised these issues in the 2nd revision. MPAA stands for mercaptophenylacetic acid and we corrected these mistakes in the 2nd version.*

Line 264-286: This paragraph doesn’t really add much to the present story. The SPAAC chemistry of azide modified proteins is fairly standard. Hence, this paragraph could and probably should be shortened.

Answer: *Thanks for suggestion. We revised this part again in the 2nd revision.*

Reviewer #2 (Remarks to the Author):

Zeng et al. addressed all the points/concerns raised by the reviewers in the revised manuscript and rebuttals. The revision has helped improve the quality of their work significantly. For example, they conducted the pH scan experiment as I suggested, and the results cleared the doubts on the optimal pH of the concerned reaction. I don't see any major issues in the revised manuscript and would recommend publication after a few minor points are taken care of.

1) Graphic abstract: One of the R groups is listed as hydrazone, but it is actually a hydrazide.

2) Legend of Figure 3: description for (c) is missing.

3) Table 1: On the Buffer column, 'pH' should be added before the numeric value in PB 6.0, PB 7.0, PB 8.0, PB 9.0. For example, PB pH 6.0.

The footnote of Table 1: aThe reactions were performed at 4 °C and corresponding buffer (50 mM, final pH=8.0 unless otherwise indicated).

4) There are still many grammatical errors in the manuscript. To give just one instance: Line 30, 1st sentence: "Chemically modification ..." should be "Chemical modification ..."

I trust the editorial office will help the authors to fix the English problems.

Answer: *Thanks for the second review and comments. According to the nice suggestions and corrections, we again revised our manuscript and all these issues have been processed in the 2nd revision version. We also tried our best to improve the writing and are going to seek the help from the editorial office during the proof step if it is accepted.*

REVIEWERS' COMMENTS

Reviewer #1 (Remarks to the Author):

The authors have satisfactorily addressed all the concerns and questions raised during the review process.